# Extracellular matrix mediates circulating tumor cell clustering in triple-negative breast cancer metastasis

Georg OM Bobkov [1,4], Khushali J. Patel [1,4], Bree M. Lege [1], Rong Zheng[1], Gad Shaulsky [2], Matthew J. Ellis[3] & Chonghui Cheng [1]✉

Metastatic tumor cell dissemination is the leading cause of cancer-related deaths. Clustered circulating tumor cells (CTCs) possess higher metastatic potential than single CTCs. Epithelial adherens junction (AJ) proteins typically mediate stable cell-cell interactions; however, these proteins are frequently lost in highly aggressive triple-negative breast cancers (TNBCs), raising the question of how CTCs from such tumors cluster. Here we show that the extracellular matrix (ECM) component hyaluronan (HA) mediates AJ-independent CTC clustering in TNBCs. HA is necessary and sufficient to drive clustering of tumor cells expressing its receptor CD44. Mechanistically, HA initiates contact between neighboring cells through actin-based membrane protrusions. As cells are pulled closer, these initial interactions expand to membrane-membrane contact and are subsequently stabilized by desmosomes. CTC-derived HA also acts as a docking platform to promote heterotypic cluster formation by recruiting non-CTCs, including immune cells. Thus, this ECM–receptor interaction enables CTC clustering and survival under shear stress, enhancing TNBC metastasis.

Metastatic spreading of tumor cells to secondary sites remains the major cause of cancer-related deaths. Circulating tumor cells (CTCs) represent an intermediate stage of metastasis and travel through the blood as single cells or as clusters. The latter can be composed solely of CTCs, or exist as heterotypic clusters that include other cell types, such as white blood cells[1–4], platelets[5,6], or cancer-associated fibroblasts[7,8]. CTC clusters can arise from collective migration of tumor cells from the primary tumor or by clustering of single CTCs near the vasculature[9–11]. Compared to single CTCs, clustered CTCs possess significantly higher metastatic potential[9], highlighting how multi-cell cooperation can overcome natural barriers against metastasis[12,13].

Previous studies have shown that stable cell-cell contacts in CTC clusters can be mediated through direct interactions between epithelial adherens junction (AJ) proteins[9,14], but these proteins are often lowly expressed or completely absent in the aggressive triple negative breast cancer (TNBC) subtype[15–18]. These TNBCs, which are highly metastatic and mesenchymal in nature, tend to express alternative adhesion proteins that primarily facilitate cell migration rather than stable cell-cell contacts[19–21]. Alternatively, TNBC CTCs have been reported to cluster through homotypic interactions of the cell surface receptor CD44[11]. However, CD44 is broadly expressed across most cell types present in the bloodstream, including immune cells, erythrocytes, and platelets[22–26]. It is thus unlikely that a clustering mechanism solely based on CD44 expression can be utilized in the bloodstream, as it would result in the formation of cell aggregates and emboli in circulation.

In addition to being mediated directly by cell surface proteins, cell-cell interactions can, in principle, also occur through extracellular matrix (ECM) components, allowing neighboring cells to interact by

[1]Department of Molecular and Human Genetics, Lester & Sue Smith Breast Center, Baylor College of Medicine, Houston, TX, USA. [2]Department of Molecular and Human Genetics, Baylor College of Medicine, Houston, TX, USA. [3]Lester and Sue Smith Breast Center and Dan L. Duncan Comprehensive Cancer Center, Baylor College of Medicine, Houston, TX, USA. [4]These authors contributed equally: Georg OM Bobkov, Khushali J. Patel. ✉e-mail: Chonghui.Cheng@BCM.edu

binding to the same ECM molecules. The ECM is a network of macromolecules that represents a key non-cellular component of tissues. It provides structural and mechanical support and acts as a dynamic signaling hub to actively shape the behavior of resident cells. Surprisingly, although CTCs do not reside in a tissue, recent studies have shown that several ECM components are upregulated in breast and pancreatic CTCs[27,28]. While extensive ECM reorganization affects every stage of the metastatic cascade[29], most studies focused on ECM remodeling at primary tumor sites and the metastatic niche[29]. Accordingly, if and how ECM components shape tumor cell behavior during circulation and whether the ECM can mediate cell-cell interactions in the bloodstream remains largely unknown.

Here, we show that cells lacking typical epithelial cell-cell adhesion molecules can utilize the ECM component hyaluronan (HA) to form stable cell-cell interactions in a cell-ECM-cell fashion. This interaction is subsequently strengthened through desmosome formation.

Furthermore, HA-based mechanism enables TNBC CTCs to form highly metastatic CTC clusters and facilitates seamless incorporation of non-tumor cells into the cluster, thereby further enhancing their metastatic potential.

## Results

### ECM-associated changes in TNBC include upregulation of *HAS2*

To determine the cellular components that drive the aggressiveness of TNBCs, we compared TNBC and non-TNBC tumors from The Cancer Genome Atlas (TCGA) breast cancer (BRCA) dataset. Gene ontology (GO) analysis revealed that ECM and plasma membrane genes are enriched in TNBCs compared to non-TNBCs (Fig. 1a), as well as in tumors expressing low levels of E-cadherin, a key component of epithelial AJs (Supplementary Fig. 1a). This suggests that altered ECM dynamics could play a role in promoting the aggressive behavior of TNBCs. Further examination of individual ECM components

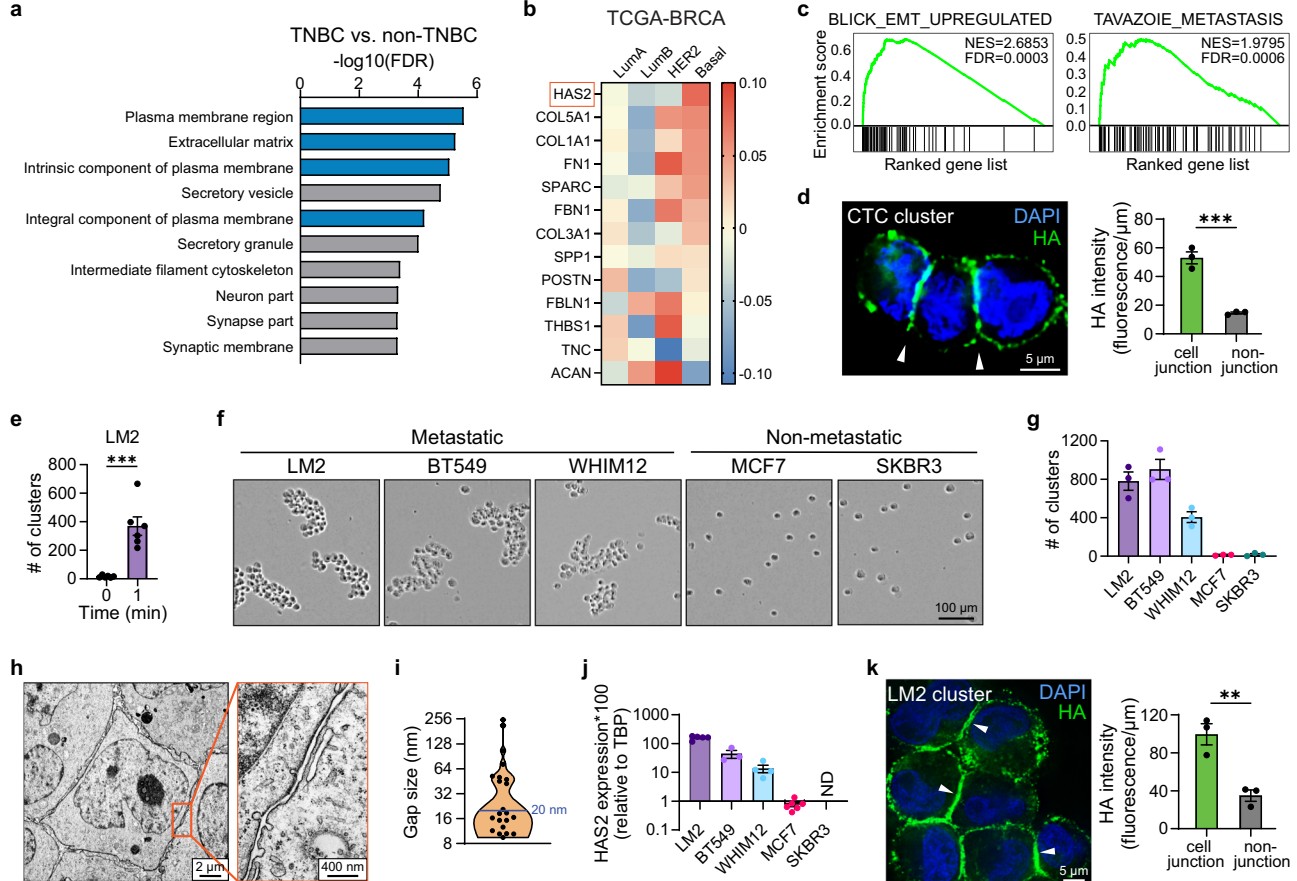

**Fig. 1 | HAS2 expression correlates with clustering of TNBC cells. a** Gene Ontology analysis showing top cellular component terms positively enriched in TNBC tumors in the TCGA-BRCA dataset. ECM and plasma membrane terms are shown in blue; other terms are shown in gray. **b** Heatmap showing expression of ECM-related genes implicated in breast cancer progression by breast cancer subtype in the TCGA-BRCA dataset. **c** GSEA plots showing significant positive enrichment of EMT and breast cancer metastasis signatures in HAS2-correlated gene list from the TCGA-BRCA dataset. **d** Left: maximum intensity projection of fixed CTC cluster isolated from LM2 tumor-bearing mice and stained for HA (green pseudo color). White arrowheads indicate HA enrichment at cell-cell interaction sites. CTCs were identified using a GFP tag. Right: quantification of HA intensity at cell-cell interaction sites ("cell junction") and cell membrane regions that are not in contact with neighboring cells ("non-junction") in CTC clusters (*n* = 3 mice; *P* = 0.0009). **e** Quantification of cluster number for LM2 cells after no clustering (0 min) or 1 min of clustering (*n* = 6 biological replicates; *P* = 0.0003). **f** Representative images of cells after 1 h of clustering. Metastatic and non-metastatic breast cell lines are

shown. **g** Quantification of cluster number for each cell line shown in **f** (*n* = 3 biological replicates). **h** Transmission electron microscopy image of clustered LM2 cells at 2000x magnification. Box indicates the position of the cell-cell interaction site displayed at 15000x magnification on the right. **i** Quantification of gap size in between interacting cells exemplified in **h** (*n* = 22 gap sizes from three biological replicates). The horizontal blue line indicates 20 nm, which is the reported gap size for adherens junctions. **j** qRT-PCR analysis of HAS2 mRNA expression in each cell line shown in (**f**). Data is normalized to TBP. ND indicates no detectable HAS2 expression (*n* = 5 (LM2), 3 (BT549), 4 (WHIM12), 6 (MCF7), 3 and (SKBR3) biological replicates). **k** Left: maximum intensity projection of a fixed in vitro LM2 cell cluster stained for HA (green). White arrowheads indicate examples of HA enrichment. Right: quantification of HA signal at cell-cell interaction sites ("cell junction") and membrane areas that are not in contact with neighboring cells ("non-junction") in LM2 clusters (*n* = 3 biological replicates; *P* = 0.0070). Data are represented as mean ± SEM. Statistical significance: **\**P* = < 0.01; \*\*\**P* = < 0.001 (unpaired two-sided *t* test). DAPI (blue) served as a nuclear counterstain.

implicated in breast cancer progression[30] showed a consistent pattern across three independent breast cancer datasets, where hyaluronan synthase 2 (*HAS2*) was one of the top two most enriched ECM-related genes in the basal-like subtype of tumors, which are predominantly TNBCs (Fig. 1b and Supplementary Fig. 1b–c).

*HAS2* encodes an integral membrane protein that synthesizes the major ECM component HA by polymerizing monosaccharides and extruding the growing polysaccharide chain directly into the extracellular space[31,32]. While HAS1 and HAS3 can also produce HA, HAS2 is the major HA-producing enzyme in breast cancer[33]. Gene set enrichment analysis (GSEA) of *HAS2*-correlated genes in TCGA-BRCA dataset revealed positive enrichment of ECM-receptor interactions, cancer-associated signaling pathways (Supplementary Fig. 1d), and aggressive tumor signatures, including epithelial-to-mesenchymal transition (EMT), breast cancer metastasis, claudin-low characteristic, and cancer stem cells (Fig. 1c and Supplementary Fig. 1e). These results are consistent with previous studies showing that elevated HA deposition or increased *HAS2* expression is associated with poor survival and metastatic status in breast cancer patients[33,34]. Despite these earlier publications, the direct mechanism by which HA promotes tumor progression has remained elusive.

Since metastases originate from CTCs, we examined the localization of HA in CTCs from the well-established metastatic TNBC mouse model MDA-MB-231-LM2 (LM2). Immunofluorescence (IF) staining revealed that HA coated the surface of LM2 CTCs, indicating that HA remains stably associated with CTCs as they travel through the bloodstream (Fig. 1d). Notably, HA specifically localized to cell-cell interaction sites within CTC clusters (Fig. 1d), raising the possibility that HA enhances the metastatic potential of CTCs by enabling stable CTC-CTC connection in clusters.

## Metastatic TNBC cells form clusters and express elevated levels of *HAS2*

To investigate whether HA indeed facilitates stable cell-cell interactions in clusters, we developed an in vitro tumor cell cluster assay to examine whether tumor cells can form clusters that remain intact while encountering physiological levels of shear stress. In this assay, freshly detached single cells are rotated on an orbital shaker at a frequency of 200 rpm (Supplementary Fig. 1f), which creates approximately 9 dyn/cm² shear stress[35]. This resembles shearing forces that CTC clusters encounter in larger arteries and smaller veins[36]. Clustering ability was assessed by quantifying the number and size of tumor cell clusters as well as the number of remaining single cells.

Exposing the metastatic TNBC cell line LM2 to this experimental setup resulted in rapid assembly of hundreds of multicellular clusters within one minute (Fig. 1e and Supplementary Fig. 1g). Circulation time of CTCs has been reported to range from minutes to several hours[37,38], which prompted us to extend the cluster assay to 1 h and examine the stability of these rapidly formed clusters. We found that LM2 tumor cell clusters remain intact even after 1 h exposure to shear stress and are bigger and tighter (Supplementary Fig. 1g). Similarly, the metastatic TNBC cell lines BT549, WHIM12, and MDA-MB-231 formed highly stable clusters when subjected to the cluster assay (Fig. 1f, g and Supplementary Fig. 1h–j). In contrast, non-metastatic breast tumor cell lines MCF7, BT474, and SKBR3, as well as the non-tumor breast cell lines HMLE and MCF10A, failed to cluster and instead remained as single cells (Fig. 1f, g and Supplementary Fig. 1h–i).

Characterization of stable LM2 tumor cell clusters via electron microscopy (EM) revealed that the cell membranes of connected cells interacted across a maximum length of 3.5 μm with a median gap of 18.6 nm between opposing membranes (Fig. 1h-i and Supplementary Fig. 1k). This gap size is similar to the 10-20 nm reported for AJs[39,40]; however, these interactions are not mediated by AJs, as the cluster-forming metastatic cell lines lack expression of the crucial AJ component E-cadherin (Supplementary Fig. 1l). Interestingly, all metastatic

tumor cells capable of clustering express *HAS2*, and its expression correlated positively with the number of clusters and negatively with the number of remaining single cells (Fig. 1j and Supplementary Fig. 1m). Together with the specific enrichment of HA at the cell-cell interaction sites inside CTC clusters (Fig. 1d), this suggests that HA produced by HAS2 mediates cell clustering.

## HA is required for CTC cluster formation

We developed an IF method to examine HA localization in three-dimensional (3D) cell clusters formed in the cluster assay. IF analysis of LM2 and WHIM12 3D tumor cell clusters revealed that HA is present on the cell surface of clustered tumor cells and is enriched at the cell-cell junctions (Fig. 1k and Supplementary Fig. 1n), which closely resembles the HA localization in CTC clusters isolated from tumor-bearing mice (Fig. 1d). Quantification of HA compared to membrane staining further confirmed the enrichment of HA at cell junctions (Supplementary Fig. 1o). These results reveal that tumor cell clusters formed in the in vitro assay recapitulate aspects of CTC clusters in vivo.

To directly test whether HA facilitates TNBC tumor cell clustering, we treated LM2 cells with the enzyme hyaluronidase (HAse) to degrade HA (Fig. 2a). In contrast to control cells, HAse-treated cells failed to form stable clusters and remained single in the cluster assay (Fig. 2b and Supplementary Fig. 2a). Similar results were obtained when HA was depleted using two independent shRNAs to knockdown (KD) *HAS2* (Supplementary Fig. 2b–e) and when treating two additional TNBC cell lines with HAse: PDX-derived WHIM12 and BT549 (Supplementary Fig. 2f). Together, these results support the notion that HA is necessary for TNBC cell clustering.

HA molecules can exist in different sizes and can elicit distinct cellular responses[33]. To assess what form of HA facilitates tumor cell clustering, we treated LM2 HAS2 KD cells for 1 h with increasing concentrations of two different sizes of HA molecules, a low molecular weight (130–150 kDa) HA and a high molecular weight (750–1000 kDa) HA. Incubation with exogenous high molecular weight HA reconstituted clustering capabilities of LM2 HAS2 KD cells in a dose-dependent manner, whereas low molecular weight HA failed to rescue clustering (Supplementary Fig. 2g). These results are consistent with HAS2's function in producing HA molecules of higher molecular weight[41] and suggest that large HA molecules are required to mediate tumor cell clustering, likely by providing enough interaction sites for HA receptors on two or more tumor cells to stably interact with the same HA moiety.

## HA-CD44 interaction is sufficient to induce tumor cell clustering

The above results suggest that metastatic tumor cells rely on HA to connect and form clusters. This prompted us to investigate whether HA binding to its cell surface receptor is required. We knocked out (KO) the major HA receptor *CD44* in LM2 cells (Supplementary Fig. 2h) and found that CD44 KO cells failed to cluster (Supplementary Fig. 2i), similar to a previous report[11]. Interestingly, the surface of CD44 KO cells is not coated by HA (Supplementary Fig. 2j), suggesting that CD44 is necessary to retain HA on the cell surface. Importantly, CD44 alone is not sufficient to drive tumor cell clustering, as the TNBC cell line SUM159, which expresses CD44 at levels comparable to that of LM2 (Supplementary Fig. 2k), failed to cluster (Fig. 2c, d). Since SUM159 cells almost completely lack *HAS2* expression (Supplementary Fig. 2l) and have no HA coat (Fig. 2e and Supplementary Fig. 2m), we ectopically expressed *HAS2* cDNA (Supplementary Fig. 2n). The HAS2-expressing SUM159 cells established an HA coat (Fig. 2f) and acquired the ability to form stable clusters under shear stress (Fig. 2g). These results suggest that HA is sufficient to induce TNBC cell clustering through its receptor CD44.

Our results predict that HA-based cell clustering is mediated through direct interactions between HA and CD44. To test this directly, we performed a reconstitution experiment using non-

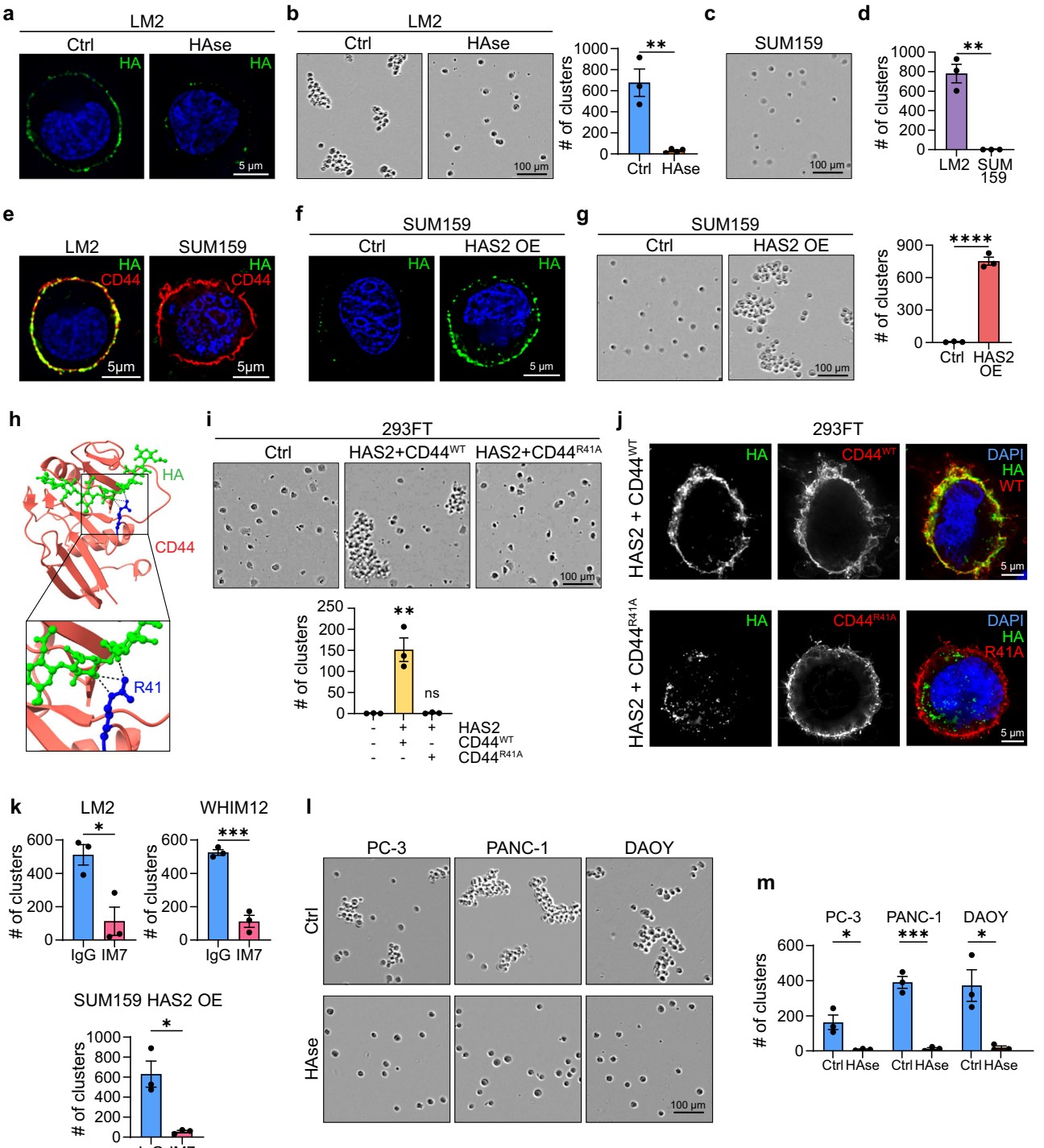

clustering 293FT cells, which lack endogenous *HAS2* and *CD44* expression (Supplementary Fig. 2o, p) and co-expressed *HAS2* with either wildtype *CD44* (*CD44^WT*) or its mutant *CD44^R41A* (Supplementary Fig. 2q, r). The CD44^R41A mutation disrupts the hydrogen bond interactions between the Arginine residue R41 of CD44 and the polysaccharide sugar chain of HA, thereby abolishing the CD44-HA interaction without affecting CD44 protein conformation[42] (Fig. 2h). 293FT cells co-expressing HAS2 and wildtype CD44, but not CD44^R41A, readily formed clusters (Fig. 2i). In contrast to the clear HA coat in CD44^WT expressing cells, only intracellular HA was detected in CD44^R41A expressing cells despite proper surface localization of CD44^R41A (Fig. 2j). These results indicate that the HA-CD44 interaction is both essential and sufficient for cluster formation. Supporting this finding,

treatment with the CD44 monoclonal antibody IM7, which blocks the interaction between HA and CD44[43], significantly reduced tumor cell clustering in LM2, WHIM12 PDX, and HAS2-overexpressing SUM159 cells (Fig. 2k). Thus, HA facilitates stable TNBC cell clustering through interactions with its receptor CD44.

To further generalize our findings in other cancer cell types, we performed the cluster assay using the prostate PC-3, the pancreatic PANC-1, and the glioblastoma DAOY cell lines. All three cell lines express *HAS2* and *CD44* at similar levels (Supplementary Fig. 2s), formed clusters under shear stress, and lost clustering ability upon HAse treatment (Fig. 2l-m). Together, these results support the notion that the HA-based clustering mechanism is employed by many metastatic cancer types in addition to TNBC cells.

**Fig. 2 | HA-CD44 interaction is necessary and sufficient for clustering.**
**a** Maximum intensity projection of LM2 cells fixed in suspension and stained for HA (green) after 1 h of Control (Ctrl) or HAse treatment. **b** Representative images and quantification of numbers of clusters formed by Ctrl and HAse-treated LM2 cells after 1 h of clustering ($n = 3$ (Ctrl) or 4 (HAse) biological replicates; $P = 0.0020$). See Supplementary Fig. 2a for larger fields of view. **c** Representative image of SUM159 cells after 1 h of clustering. **d** Quantification of the number of clusters formed by LM2 and SUM159 cells after 1 h of clustering ($n = 3$ biological replicates; $P = 0.0012$). **e** Maximum intensity projection of LM2 and SUM159 cells fixed in suspension and stained for HA (green) and CD44 (red). See Supplementary Fig. 2m for single channel images. **f** Maximum intensity projection of SUM159 Ctrl and HAS2 OE cells fixed in suspension and stained for HA (green). **g** Representative images and quantification of numbers of clusters formed by SUM159 Ctrl and HAS2 OE cells after 1 h of clustering ($n = 3$ biological replicates). **h** Crystal structure of the interaction between HA (green) and mouse CD44 HA binding domain (red). Dotted black lines represent hydrogen bonds. The CD44 residue corresponding to human

R41 is highlighted in blue. Box indicates the position of the magnified region below. Generated using the structure deposited at Protein Data Bank ID 2JCR and the UCSF ChimeraX software. **i, j** Results for 293FT control and 293FT co-transfected with either HAS2 + CD44[WT] or HAS2 + CD44[R41A] are shown. Representative images and quantification of numbers of clusters formed by 293FT cells after 1 h of clustering ($n = 3$ biological replicates; $P = 0.0014, 0.9990$) (**i**). Maximum intensity projection of freshly detached cells stained for HA (green) and CD44 (red) (**j**). **k** Quantification of the number of clusters formed by IgG- or CD44 blocking antibody IM7-treated LM2, WHIM12, and SUM159 HAS2 OE cells after 1 h of clustering ($n = 3$ biological replicates; $P = 0.0189, 0.0005, 0.0118$). Representative images (**l**) and quantification (**m**) of the number of clusters formed by control and HAse-treated non-breast cancer cell lines ($n = 3$ biological replicates; $P = 0.0197, 0.0004, 0.0170$). Data are represented as mean ± SEM. Statistical significance: ns = not significant; *$P = < 0.05$; **$P = < 0.01$; ***$P = < 0.001$; ****$P = < 0.0001$ (unpaired two-sided $t$ test (**b, d, g, k, m**) or ordinary one-way ANOVA (**i**)). DAPI (blue) served as a nuclear counterstain.

## Actin-based cellular protrusions are required for tumor cell clustering

IF analysis of LM2 cells in 3D suspension revealed numerous CD44-positive cellular protrusions. These protrusions radiated outwards from the main cell body with a maximum length of 0.25-0.3 μm, showed a specific enrichment of HA at their tips (Fig. 3a), and their lengths were not affected by HA levels or CD44 expression (Fig. 3b). By greatly expanding the cell surface area, these protrusions likely serve as initial points of contact between neighboring cells, and HA's localization at their distal tips optimally positions it to initiate cellular interactions. Indeed, when analyzing early cell-cell interaction events in freshly detached cells, we observed that cells formed connections via protrusions that were tightly associated with HA across their full length (Fig. 3c). Interestingly, CD44 and HA staining of neighboring cells grown in 2D also revealed interacting protrusions (Supplementary Fig. 3a).

To understand the cytoskeletal basis of these protrusions, we co-stained LM2 cells in suspension for CD44 and two major cytoskeletal components, actin and tubulin. Actin staining showed a clear overlap with the CD44-marked protrusions (Fig. 3d), whereas tubulin localized just beneath the cell membrane (Supplementary Fig. 3b). Treatment of LM2 cells with the actin depolymerizing drug Latrunculin A, but not the microtubule polymerization inhibitor colchicine, eliminated protrusions[44] and increased cell size[45] (Supplementary Fig. 3c). In addition, cell surface staining of CD44, and especially HA, was markedly reduced in Latrunculin A treated cells (Supplementary Fig. 3c) suggesting that actin is required to properly position CD44, and thereby also HA at the cell surface. Consistent with HA being required for TNBC tumor cell clustering (Fig. 2 and Supplementary Fig. 2), Latrunculin A treatment abolished cluster formation, whereas clustering remained largely unaffected by colchicine (Fig. 3e and Supplementary Fig. 3d). Together, these results suggest that actin is essential for the formation of tumor cell clusters, likely through the formation of cellular protrusions that help organize HA localization (Figs. 2j and 3a).

## Cellular protrusions facilitate HA-dependent tumor cell clustering

To investigate whether cellular protrusions play a physical role in the initiation and maturation of cell contacts, we analyzed early cell clustering events using live cell imaging. We found that neighboring cells initiate contact through cellular protrusions, and they subsequently move closer to each other, gradually expanding their contact interface (Fig. 3f, Supplementary Fig. 3e and Supplementary Movies 1–3). Within 30–45 min, this process resulted in the formation of a mature connection, with cells adhering across a large membrane area, similar to the cell-cell interaction sites observed in the EM images (Fig. 1h) and CTC clusters (Fig. 1d).

We also conducted live imaging of mixed LM2 control and CD44 KO cells, as the latter fail to establish an HA coat (Supplementary Fig. 2j). We labeled actin to visualize protrusions and CD44 to distinguish between control (yellow) and CD44 KO (red) cells. Protrusions of neighboring control cells readily initiated contact and ultimately formed tight clusters (Fig. 3g, cyan arrows and Supplementary Movie 4). In contrast, CD44 KO cells failed to establish stable cell contacts with either control or other CD44 KO cells despite being in close proximity (Fig. 3g, yellow arrows and Supplementary Movie 4). During up to 2.5 h of live imaging, the distance between neighboring control cells rapidly decreased as they clustered, whereas neighboring KO cells remained separated by at least 0.5 μm (Fig. 3h), which corresponds to approximately twice the average length of the longest protrusions (Fig. 3b). Collectively, these results suggest that the HA-CD44 interaction is crucial to connect actin-based protrusions of neighboring cells and ultimately enables the formation of tumor cell clusters.

## Sliding of protrusions alongside each other enables maturation of cell connections

The above data suggests that once cell-cell contact is initiated at the level of protrusions, cells are gradually pulled closer until they form stable interactions across a large cell surface area (Fig. 3f–h, Supplementary Fig. 3e and Supplementary Movies 1–4). This maturation process could be a consequence of retraction of Head-to-Head connected protrusions, or the result of attached protrusions sliding alongside each other until they finally connect to the opposite cell body. To evaluate these possibilities, we analyzed protrusion attachment types in early cell-cell interactions using EM (Fig. 3i–k and Supplementary Fig. 3f). We observed a clear relationship between the distance separating neighboring cells and the number of their cell-cell connections (Fig. 3j), suggesting that once initial contact is established by a few long protrusions, additional protrusions are progressively recruited during the maturation process.

We identified five categories of protrusion interactions (Fig. 3i and Supplementary Fig. 3f). (1) Head-to-Head, (2) Head-to-Side, (3) Side-to-Side, (4) direct protrusion-membrane interaction, and (5) broad membrane-membrane contacts. Quantifying these interaction types relative to the distance between cells revealed a clear pattern. Long-distance interactions were predominantly Head-to-Head, which transitioned to Head-to-Side and Side-to-Side interactions as cells drew closer, finally culminating in direct membrane-membrane contacts (Fig. 3k). These observations indicate that cells progressively increase their contact area during the maturation of the cell-cell interaction site, ultimately resulting in direct membrane-membrane contacts. These results suggest that the maturation process involves protrusion-guided sliding of cells closer to each other, and not retraction of Head-to-Head connected protrusions. Intriguingly, this HA-mediated

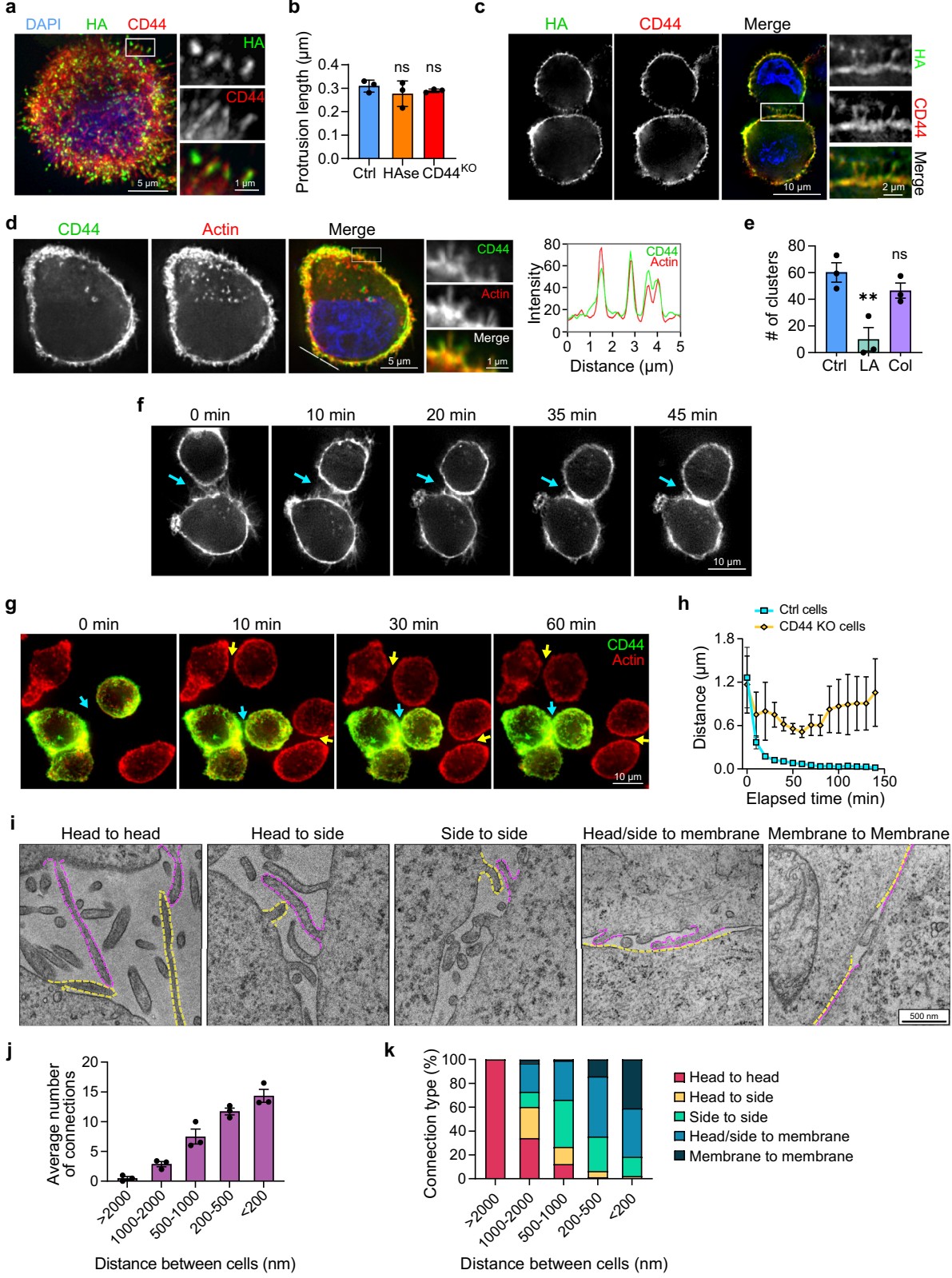

clustering mechanism parallels the E-cadherin-mediated AJ formation process[46,47] (see "Discussion").

## HA promotes cluster stability

The above analysis of the maturation process revealed that cells strive to maximize the surface contact area between neighbors to form stable HA-based cell-cell connections. As shown in Fig. 2, HA is required for tumor cells to establish stable multicellular clusters under shear stress. However, HA has previously been reported to be dispensable for tumor cell aggregation in an assay where cells settle in a low-attachment plate[11]. These seemingly contradictory findings can be reconciled if HA is dispensable for unperturbed cluster formation but required for stable clusters that are capable of withstanding physiological levels of shear stress.

**Fig. 3 | HA-mediated clustering is initiated by actin-based protrusions. a** 3D projection of a freshly detached LM2 cell fixed in suspension. Box indicates the position of the 3-fold magnified region on the right which shows the localization of HA at the tips of protrusions. Cells were labeled for HA (green) and CD44 (red). **b** Quantification of longest protrusion length in LM2 Ctrl, HAse-treated and CD44KO cells (n = 3 biological replicates; P = 0.4640, 0.7294). Protrusion length was quantified from actin-labelled protrusions. **c** Maximum intensity projection of fixed early-interacting LM2 cells stained for HA (green) and CD44 (red). Box indicates the position of the 2-fold magnified region on the right demonstrating HA localization at sites of interacting protrusions. **d** Left: single optical section of a detached LM2 cell stained for CD44 (green) and actin (red). Box indicates the position of the 3-fold magnified region on the right, showing protrusions. Line indicates position of the line profile (right) for CD44 (green) and actin (red) signals in protrusion. See Supplementary Fig. 3b for tubulin staining of the same cell. **e** Quantification of the number of clusters formed by LM2 cells treated with Latrunculin A (LA) or Colchicine (Col) after 1 h of clustering (n = 3 biological replicates; P = 0.0052, 0.3744). See Supplementary Fig. 3d for number of remaining single cells. **f** Live imaging of early interaction events between freshly detached LM2 cells. Single optical sections are shown; cellular protrusions were visualized using

anti-CD44 staining. Minutes that passed since the start of the experiment are displayed above each picture. See Supplementary Movies 1–3 and Supplementary Fig. 3e for another example of the maturation process. **g** Live imaging of freshly detached LM2 Ctrl and LM2 CD44KO cells. Cells were labeled for CD44 (green) and actin (red). Arrows indicate interactions between Ctrl cells (cyan) or KO cells (yellow). Minutes that passed since the start of the experiment are displayed above each picture. See Supplementary Movie 4. **h** Quantification of live imaging experiment shown in (**g**). Distance between Ctrl cells (cyan) or KO cells (yellow) over time is displayed (n = 3 biological replicates). **i** Transmission electron microscopy pictures depicting the five types of interactions observed in LM2 WT cells after clustering for 5–20 min. Images were taken at 5000–15000x magnification. Purple and yellow dotted lines indicate membrane regions involved in each interaction. See also Supplementary Fig. 3f. **j** Number of connections between cells exemplified in (**i**). Results were categorized by distance between interacting cells (n = 3 biological replicates). **k** Quantification of the cell-cell interaction types depicted in (**i**) Interactions were categorized by distance between interacting cells. Data are represented as mean ± SEM. Statistical significance: ns = not significant; **P = < 0.01 (ordinary one-way ANOVA). DAPI (blue) served as a nuclear counterstain.

To test this, we formed tumor cell aggregates of LM2 control and HAse-treated cells in a low-attachment plate and subsequently subjected them to increasing amounts of shear stress (Fig. 4a). In agreement with the previous report[11], LM2 control and HAse-treated cells showed similar numbers of clusters and single cells in the absence of shear stress (Fig. 4a and Supplementary Fig. 4a–c, "no shear stress"). However, cell aggregates formed in the absence of HA were only stable at very low rotational speeds. When subjected to physiological levels of shear stress, HAse-treated aggregates rapidly dissociated into single cells, while control clusters remained largely intact, supporting the notion that HA is required to form stable connections (Fig. 4a and Supplementary Fig. 4a–c, "shear stress"). Similarly, WHIM12 PDX tumor cells required HA to form functional cell clusters capable of withstanding shear stress (Supplementary Fig. 4d–f).

To gain mechanistic insight into how HA stabilizes cell clusters, we next used EM to compare shear stress-free cell aggregates formed in the presence or absence of HA. Cell-free spaces in HAse-treated LM2 cell aggregates were approximately two-fold higher compared to the LM2 control aggregates (Fig. 4b, c and Supplementary Fig. 4g), indicating that cells in HAse-treated aggregates are only loosely associated. Moreover, the interconnectivity between cells in aggregates formed in the absence of HA was limited and control cell aggregates formed a network that was significantly more densely connected (Figs. 4b, d and Supplementary Fig. 4g). Tightly packed cells in control aggregates were mostly cobblestone shaped, whereas HAse-treated cells retained the predominantly round morphology of single cells (Fig. 4b and Supplementary Fig. 4g). Collectively, these findings suggest that HA is required to form stable cell clusters that can withstand physiological levels of shear stress. This stability is achieved by strengthening individual cell-cell contacts through interactions across a larger membrane surface area and by increasing the overall interconnectivity of cells within the cluster. These results show that while HA is dispensable for TNBC cell cluster formation in the absence of shear stress, it is critically required to form functional tumor cell clusters that can withstand the shear stress encountered by CTC clusters in the bloodstream.

Examining the E-cadherin-positive MCF7 cell lines revealed that although they failed to cluster under shear stress (Fig. 1f, g), MCF7 cells can efficiently form clusters when left unperturbed in a low-attachment plate (Supplementary Fig. 4h). Subsequent addition of shear stress did not break apart the clusters, in line with the reported strength of AJ-dependent cell-cell interactions[48] (Supplementary Fig. 4h). This suggests that AJ-dependent clusters likely form under no-shear stress conditions in the tumor bed and shed as intact clusters.

## Desmosomes stabilize HA-mediated clusters

The tight interactions observed in fully formed clusters prompted us to investigate whether HA-based clustering recruits additional components to strengthen the cell-cell connection during the maturation of the interaction site. Desmosomes are specialized intercellular junctions that connect to intermediate filaments to stably link neighboring cells together[49]. Desmoglein-2 (DSG2) and Desmocollin-2 (DSC2) are desmosomal cadherins that mediate adhesive binding between adjacent cells, while Desmoplakin (DSP) is the intracellular scaffold protein that anchors these cadherins to intermediate filament of the cytoskeleton. Interestingly, recent reports showed that elevated DSG2 expression increased prevalence of CTC clusters and metastasis[50] and that DSC2 and DSP were associated with increased cluster formation and CTC survival in breast and lung cancer cells[51].

These findings prompted us to investigate whether desmosomes are involved in HA-based clustering of tumor cells as LM2 cells express DSG2, DSC2, and DSP (Supplementary Fig. 4i). Assessing their localization in clustered LM2 cells, we found all three desmosome proteins localized to the fully established cell-cell interaction sites, indicating that they mechanically couple neighboring cells in mature clusters (Fig. 4e). In contrast, in early cell-cell interactions where cell morphology is still largely round and cell membrane contacts are limited to a small area, only HA, but not DSG2, DSC2 or DSP, is enriched at the cell contact site (Supplementary Fig. 4j). This suggests that desmosomes are established after large areas of the cell membranes are brought into close proximity via HA-based interactions.

To determine whether desmosomes promote HA-based clustering by strengthening cell-cell interactions, we knocked down DSG2, DSC2, and DSP individually in LM2 cells (Supplementary Fig. 4i). Next, we performed cell clustering experiments with gradually increasing levels of rotational speed. All three KD cell lines formed clusters at 200 rpm, but they rapidly dissociated at higher shear forces, indicating reduced stability compared to control cell clusters (Fig. 4f, g). Of note, control cell clusters started to break into smaller clusters at ~350 rpm (Fig. 4f), which was not the case when cell-cell interactions were established in the absence of shear stress over a period of 24 h (Fig. 4a). Taken together, these results suggest that HA-mediated initial cell contacts enable the subsequent assembly of desmosomes, which effectively lock in these interactions and mechanically stabilize the clusters.

## HA-based clustering increases cell survival under shear stress

Shear stress not only challenges cluster stability but also poses a threat for CTC survival. To examine the impact of HA on tumor cell survival

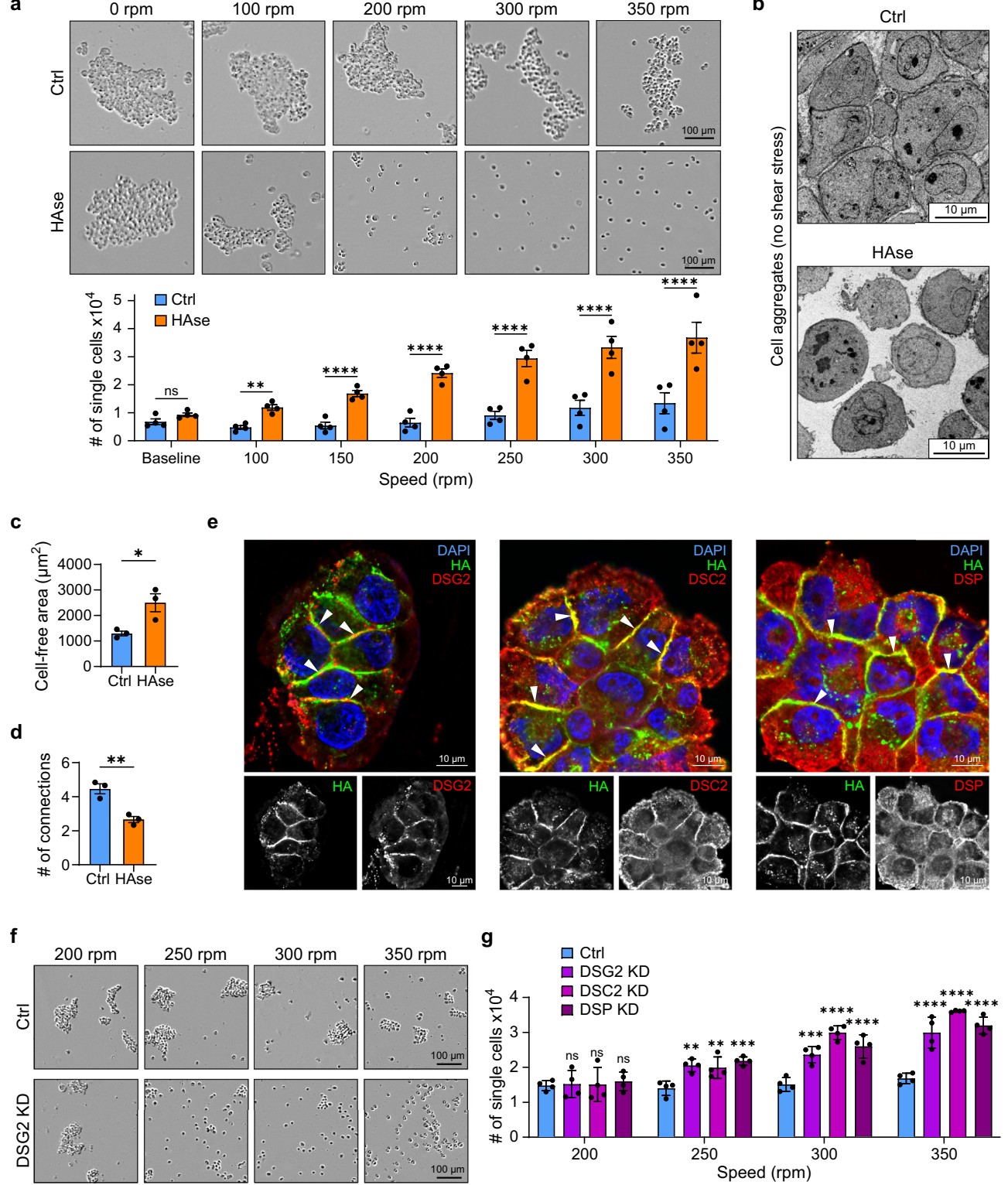

**Fig. 4 | HA and desmosomes promote cluster stability. a** Representative images (top) and quantification of the number of single cells (bottom) in Ctrl and HAse treated LM2 aggregates after shear stress exposure at various rotational speeds (*n* = 4 biological replicates; *P* = 0.2731, 0.0048). Cells were settled for 24 h in a low-attachment plate (0 rpm) and then sequentially agitated at each speed for 15 min. **b** Transmission electron microscopy images of Ctrl and HAse treated LM2 cells settled for 16 h in a low-attachment plate. Images were taken at 500x magnification. See also Supplementary Fig. 4g. Quantification of cell-free area per image (**c**; *P* = 0.0295) and average number of connections to neighboring cells per analyzed image (**d**; *P* = 0.0061) of clusters exemplified in **b** (*n* = 3 biological replicates). **e** Maximum intensity projections of fixed LM2 clusters stained with HA (green) and

DSG2, DSC2 or DSP (red). Bottom panels depict grayscale images. White arrowheads indicate enrichment at the interaction sites. See also Supplementary Fig. 4j for examples of early clusters. Representative images (**f**) and quantification of number of single cells (**g**) in LM2 Ctrl and DSG2, DSC2, or DSP KD cell clusters after shear stress exposure at various rotational speeds (*n* = 3 biological replicates; *P* = 0.9911, 0.9959, 0.8673, 0.0038, 0.0099, 0.0005, 0.0002). Cells were allowed to form clusters at 200 rpm for 30 min, after which the clusters were sequentially agitated at each speed for 15 min. Data are represented as mean ± SEM. Statistical significance: ns = not significant; *P* = < 0.05; **P* = < 0.01; ***P* = < 0.001; ****P* = < 0.0001 (Two-way ANOVA (**a**, **g**) or unpaired two-sided *t* test (**c**, **d**)).

under shear stress, we analyzed cell death in a time course experiment. HA depletion decreased cell survival, showing significant differences from 16 h onwards. At 24 h, only 20–25% of the HAse-treated or HAS2 KD LM2 cells were alive, compared to 65% of the control cells (Supplementary Fig. 5a–c). Similarly, HAse treatment reduced survival of WHIM12 PDX cells (Supplementary Fig. 5b, *right*), suggesting that HA-mediated clustering reduces shear force-induced cell death.

Shear stress induces the production of reactive oxygen species (ROS), which in turn triggers cell death[52–55]. Since cell clustering can limit ROS production[56], we tested whether oxidative stress contributes to the increased shear stress-induced cell death in HA-lacking cells. Compared to untreated wild type cells, shear stress exposure elevated ROS levels in both control and HAse-treated LM2 cells at 12 h (Supplementary Fig. 6a, b), a timepoint preceding extensive cell death of HAse treated cells (Supplementary Fig. 5a). However, the ROS levels were more than two-fold higher in single HAse-treated cells compared to clustered control (Supplementary Fig. 6a), suggesting that HA-mediated clustering promotes CTC survival by limiting shear stress-induced ROS production. Indeed, addition of the antioxidant N-acetylcysteine (NAC) to repress ROS partially rescued cell death of HAse-treated single cells (Supplementary Fig. 6c). These results indicate that HA-mediated clustering at least partially shields tumor cells from shear stress-induced cell death.

## HA is required for CTC clustering and metastasis in vivo

Increased HA synthesis[57], *HAS2* expression[58], and cell-surface bound HA[59] have all been linked to increased metastatic potential in breast and melanoma cancers, whereas loss of HAS2 is correlated with reduced metastasis[34,60]. Despite these observations, the direct mechanism of how lack of HA production negatively impacts the complex metastatic process remains elusive. Our finding that HA is required for the formation of stable TNBC cell clusters has therefore the potential to provide a mechanistic explanation for these observations.

Based on our in vitro results, we hypothesized that HA-lacking CTCs may have reduced metastatic colonization ability as they are more susceptible to shear stress-induced cell death (Supplementary Figs. 5–6). To test this hypothesis, we injected LM2 control and HAS2 KD cells via tail vein in female NSG mice and examined their lungs for the presence of GFP-positive tumor cells 24 h later. Indeed, HAS2 KD tumor cells were present in fewer numbers in the lungs compared to control cells (Supplementary Fig. 7a). Additionally, a Matrigel-based in vitro invasion assay revealed that HA-lacking single cells have reduced invasive capacity compared to clustered control cells (Supplementary Fig. 7b). Together, these results indicate that HA-mediated clustering protects tumor cells from shear-stress induced cell death and promotes invasion to enhance metastatic colonization and outgrowth.

To determine the role of HA in spontaneous CTC clustering and metastasis in vivo, we injected LM2 control and HAS2 KD cells into the mammary fat pad of female NSG mice (Fig. 5a). Tumors were collected 8 weeks after injection and continued *HAS2* KD was verified (Supplementary Fig. 7c). Histological analysis of the lungs revealed that HAS2 KD led to a more than 5-fold decrease in lung metastasis (Fig. 5b, c). Importantly, this difference is not a consequence of reduced tumor outgrowth as HAS2 KD did not significantly alter primary tumor growth in mice (Fig. 5d–e and Supplementary Fig. 7d), consistent with their similar in vitro growth rates (Supplementary Fig. 7e) and an unaltered Ki67-based proliferation index in vivo (Supplementary Fig. 7f). Analysis of blood samples (Fig. 5f) showed that HAS2 KD resulted in a drastic reduction in total CTC numbers (Fig. 5g), with 75-fold and 82-fold reduction of single and clustered CTCs, respectively (Fig. 5h and Supplementary Fig. 7g). CTC clusters were detected in all 5 control mice, compared to only 4 out of 7 mice in the HAS2 KD group. Moreover, while control CTC clusters ranged in size from 2 to 22 CTCs

per cluster, all observed CTC clusters in HAS2 KD mice were limited to only 2 CTCs per cluster (Fig. 5i). IF staining of CTC clusters from control mice revealed strong HA accumulation at CTC interaction sites (Fig. 5j and Supplementary Fig. 7h, i), consistent with our previous observations (Figs. 1d, k and Supplementary Fig. 1n, o). These results indicate that HA is essential for CTC clustering, which in turn enhances TNBC metastatic potential.

To determine whether the reduced metastasis of HAS2 KD TNBC cells is a consequence of their inability to form functional HA-mediated CTC clusters, we utilized the experimental metastasis model by injecting LM2 control and HAS2 KD cells via the tail vein. By avoiding potential confounding effects of HAS2 KD on the primary tumor microenvironment, tumor cell migration, and intravasation, this experiment allowed us to assess the stability of tumor cell clusters in the bloodstream more directly. LM2 control and HAS2 KD cells were allowed to form aggregates overnight without shear force and similar-sized clusters containing 5–10 tumor cells each were injected into the tail veins of female NSG mice (Fig. 5k). While similar numbers of both control and HAS2 KD cells reached the lung vasculature upon injection (Fig. 5k, l Day 0), clear differences in metastatic lung signal are detectable from one week onwards (Fig. 5k, l and Supplementary Fig. 7j). At the experimental endpoint of 4 weeks, the HAS2 KD group exhibited a significant decrease in lung metastasis by more than 4-fold. Histological analysis of the lungs revealed reduced tumor burden in the HAS2 KD group (Fig. 5m), consistent with our findings from the spontaneous metastasis model (Fig. 5b, c). Ki67 staining of the metastatic lungs was the same between LM2 control and HAS2 KD tumor cells (Supplementary Fig. 7k), indicating that the decreased lung metastasis was not a consequence of altered tumor cell growth. Instead, as shown earlier (Supplementary Figs. 5 and 6), the HAS2 KD cells likely arrived in the lungs in a worse condition and failed to invade (Supplementary Fig. 7a, b), resulting in reduced initial colonization. Analysis of blood samples showed that CTCs were present in 6 out of 7 control mice, but only in 2 out of 7 in the HAS2 KD group (Supplementary Fig. 7l), with one of the two KD mice having barely any CTCs (Fig. 5n). HAS2 KD resulted in greater than a 100-fold reduction in CTC numbers (Fig. 5n), including both clustered and single CTCs (Fig. 5o and Supplementary Fig. 7m). Consistent with the spontaneous metastasis experiment, HAS2 KD drastically decreased CTC cluster numbers and size. Control CTC clusters were detected in 6 out of 7 mice, with the cluster size ranging from 2 to 18 CTCs per cluster. In contrast, only 1 out of 7 mice in HAS2 KD group had clusters, which were limited in size to 2–4 CTCs per cluster (Fig. 5p). Noticeably, a small amount of HA was detectable at cell-cell interaction sites in HAS2 KD CTC clusters (Supplementary Fig. 7n), suggesting that their formation could be mediated by residual HA.

The significance of HA in CTC clustering was further supported by IF staining of CTCs from control mice. Several key features identified in our in vitro cluster assay (Figs. 2 and 3) were recapitulated in these CTC clusters: HA was enriched at the cell-cell interaction sites (Fig. 5q, r), and we detected both early interaction events involving HA-tipped protrusions (Fig. 5q, Supplementary Fig. 7o, p and Supplementary Movie 5) as well as fully matured connections in large clusters (Fig. 5r and Supplementary Fig. 7q). Looking at early CTC clustering in 3D revealed two cells connected at a small membrane region, which was surrounded by interlocked protrusions of varying lengths. Reminiscent of our in vitro results (Fig. 3f–k and Supplementary Fig. 3e, f), these protrusions appeared to be sliding alongside each other to bring both CTCs closer together and expand and stabilize the contacting area (Supplementary Movie 5). As HA localized to both interacting protrusions and the directly contacting areas of CTCs, these results suggest that HA enhances TNBC metastasis in vivo by enabling the formation of stable CTC clusters.

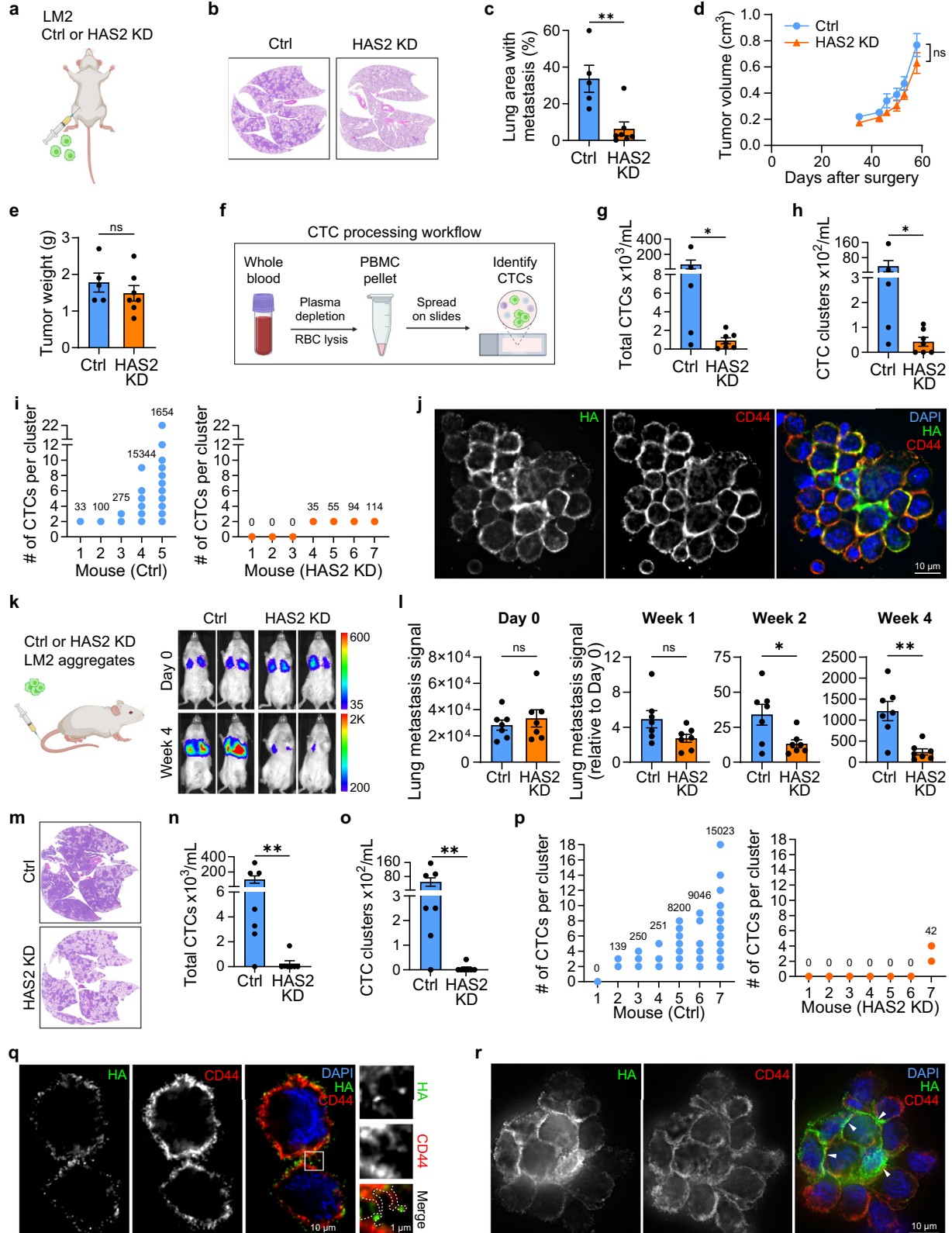

## HA facilitates interactions between CTCs and non-CTCs

Tumor heterogeneity is a key feature that enables tumors to adapt to new environments and develop therapeutic resistance[61]. Accordingly, CTC clusters can be composed of highly diverse CTCs. Although HA-deficient cells are unable to cluster on their own, the HA coat of one CTC could, in principle, serve as a docking site to recruit HA-lacking cells that express an HA receptor. Such a mechanism would enable the

formation of heterogeneous CTC clusters, where HA-producing and HA-lacking tumor cells combine their respective capabilities to increase the metastatic potential of the whole CTC cluster.

To test this hypothesis, we mixed RFP-tagged LM2 cells with GFP-tagged HAS2 KD LM2 cells in a 1:1 ratio. Even though HAS2 KD cells were unable to cluster on their own, 60% of them readily integrated into clusters with control cells (Fig. 6a and Supplementary Fig. 8a). In a

**Fig. 5 | HA is required for CTC clustering and metastasis in vivo. a** Schematic depicting mammary fat pad injection of GFP-tagged LM2 control ($n = 5$ mice) or HAS2 KD ($n = 7$ mice) single cells in immunocompromised mice. These mouse samples were analyzed in experiments (**b–j**). Representative images (**b**) and quantification (**c**; $P = 0.0049$) of H&E-stained lungs showing metastatic tumor burden at experimental endpoint. **d** Average primary tumor volume for these mice over 58 days ($P = 0.2785$). **e** Primary tumor weight at experimental endpoint ($P = 0.3963$). See Supplementary Fig. 7d for tumor pictures. **f** Schematic demonstrating the blood processing steps for CTC analysis. Quantification of the number of total CTCs (**g**; $P = 0.0048$) or CTC clusters (**h**; $P = 0.0455$) detected per mL of blood. See Supplementary Fig. 7g for number of single CTCs. **i**, Dot plots showing the unique cluster sizes observed in each mouse in control (left) and HAS2 KD (right) groups. A value of zero indicates no CTC clusters were detected. Numbers inside the graph indicate total detected clusters. **j** Maximum intensity projection of a fixed LM2 CTC cluster stained for HA (green pseudo color) and CD44 (red). See Supplementary Fig. 7h for CTC staining. **k** Left: schematic depicting tail vein injection of GFP-tagged LM2 control or HAS2 KD cell aggregates in immunocompromised mice ($n = 7$ mice per group). These mouse samples were analyzed in experiments (**l–r**). Right: representative images of lung tumor burden measured via bioluminescent imaging in these mice at time of injection (Day 0) and upon experimental endpoint at Week 4. **l** Quantification of lung metastasis signal at time of injection (Day 0; $P = 0.5176$) and Weeks 1 ($P = 0.0673$), 2 ($P = 0.0229$), and 4 ($P = 0.0016$). Data is normalized to lung signal at time of injection. See Supplementary Fig. 7j for a growth curve. **m**, Representative images of H&E-stained lungs

showing tumor burden at Week 4. Quantification of the number of total CTCs (**n**; $P = 0.0064$) or CTC clusters (**o**; $P = 0.0047$) detected per mL of blood. See Supplementary Fig. 7l–m for percent of mice with CTCs and number of single CTCs. **p** Dot plots showing the unique cluster sizes observed in each mouse in control (left) and HAS2 KD (right) groups. A value of zero indicates no CTC clusters were detected. Numbers inside the graph indicate total detected clusters. **q** Single optical section of an early connection site in a LM2 CTC cluster stained for HA (green pseudo color) and CD44 (red). Box indicates the position of the 3-fold magnified region on the right which illustrates the HA-mediated connection of protrusions (outlined with dotted white lines) from adjacent cells. See Supplementary Fig. 7o for CTC staining and Supplementary Movie 5 for a 3D depiction of the whole interaction site. See Supplementary Fig. 7p for an alternative z-panel. **r** Maximum intensity projection of a large LM2 CTC cluster stained for HA (green pseudo color) and CD44 (red). White arrowheads indicate examples of HA enrichment at cell-cell interaction sites. See Supplementary Fig. 7q for CTC staining. Data are represented as mean ± SEM. Statistical significance: ns = not significant; *$P < 0.05$; **$P < 0.01$ (unpaired two-sided $t$ test (**c–e**, **l**) or two-sided Mann-Whitney test (**g–h**, **n–o**)). Blood from 3 different Ctrl mice was analyzed to obtain the representative microscopy pictures shown in (**j**, **q**, **r**). DAPI (blue) served as a nuclear counterstain. Note: for visualization purposes, HA staining was assigned a green pseudo-color to match the color scheme used throughout the manuscript. **a**, **f**, **k** were created in BioRender. Bobkov, G. (2026) https://BioRender.com/srzuwrk.

similar experimental setup, LM2 CD44 KO cells failed to integrate into control cell clusters (Fig. 6b and Supplementary Fig. 8b), emphasizing the necessity of an HA receptor for inclusion into HA-mediated clusters. IF analysis of early cell-cell interactions between LM2 control and HAS2 KD cells revealed a partial translocation of HA from the control cells towards the HAS2 KD cells (Fig. 6c and Supplementary Fig. 8c). Upon full maturation of the cell-cell interaction site in mixed clusters, an enrichment of HA from control cells at the interaction sites between control and HAS2 KD cells is apparent (Fig. 6d and Supplementary Fig. 8d). These results reveal that the HA coat presented by CTCs can recruit HA-lacking CTCs that express the HA receptor CD44 into CTC clusters.

To evaluate this process in vivo, we pre-formed cell aggregates of RFP-tagged control and GFP-tagged HAS2 KD LM2 cells mixed at a 1:1 ratio and injected them into the tail vein of female NSG mice (Fig. 6e). After 4 weeks, we analyzed the number of CTCs originating from control and HAS2 KD cells in each mouse and observed a 4- to 12-fold reduction of CTCs originating from HAS2 KD cells (Fig. 6f). Furthermore, 64.4% of CTC clusters were composed solely of control LM2 cells, 31.4% contained a mixture of both control and HAS2 KD cells, and the remaining 4.2% consisted exclusively of HAS2 KD cells (Fig. 6g). Analysis of the individual mixed clusters revealed that the majority of CTCs in each mixed cluster were control cells (Fig. 6h). In accordance with our observations for mixed in vitro clusters (Fig. 6d), HA derived from control CTCs was enriched at their cell interaction sites where they connected to HAS2 KD CTCs (Fig. 6i).

We next analyzed HA localization in blood specimens from five TNBC patients. CTCs from patient blood were identified using a cocktail of the gold standard CTC markers (Pan-Cytokeratin, EpCAM, and EGFR). CTC clusters were detected in three of the TNBC patients. These CTCs displayed a prominent HA coat, and a clear HA enrichment was found at CTC-CTC contact sites, both in clusters where all CTCs produced HA (Supplementary Fig. 8e) and when only one CTC provided the HA necessary for the interaction (Fig. 6j). One TNBC patient had HA-coated CTCs, but no CTC clusters were detected, and the last TNBC patient only had single CTCs without an HA coat. These results reiterate the importance of HA in TNBC CTC clustering and also show that HA-coated CTCs are capable of recruiting HA-lacking CTCs into CTC clusters.

In addition to heterogeneous CTC clusters, we also observed HA enrichment in heterotypic clusters consisting of CTCs and non-CTCs in

mice (Fig. 6k–m and Supplementary Fig. 8f–i). CD44 staining revealed that a small number of protrusions from non-CTCs interacted with HA-tipped CTC protrusions during early interactions (Fig. 6k, *left*, Fig. 6l and Supplementary Fig. 8f). This example further supports the notion that neighboring cells can attach to CTCs by binding to the same HA moiety. The distinct localization pattern of HA often shifted from a widespread HA coat covering an unattached single cell (Fig. 2a) to a concentrated accumulation at the cell-cell interaction sites (Supplementary Fig. 8g), indicating that CTC-produced HA mediates the interactions with non-CTCs. Some of these non-CTCs expressed the mouse neutrophil marker Ly6G (Fig. 6k, *right*, 6m and Supplementary Fig. 8h, i), indicating that neutrophils, which are known to be part of heterotypic CTC clusters[3], can be recruited to CTC clusters via HA. Moreover, HA enrichment at interaction sites between CTCs and non-CTCs was also observed in metastatic TNBC patient blood specimens (Fig. 6n). Thus, HA-based clustering can facilitate the formation of heterotypic CTC clusters.

## Discussion

In this study, we reveal that TNBC CTCs travel in the bloodstream with a coat of the ECM molecule HA. Through a combination of in vitro assays and in vivo metastasis models, we demonstrate that this HA coat serves as the foundation for stable cell-ECM-cell interactions that drive the formation and maintenance of AJ-independent CTC clusters (Fig. 7). Mechanistically, HA initiates CTC clustering via actin-based protrusions and subsequently stabilizes mature cell-cell interactions, which are further strengthened through the assembly of desmosomes. Using TNBC patient samples and mouse models, we also show that HA can serve as a landing pad to enable integration of non-tumor cells that express the HA receptor CD44 into heterotypic CTC clusters. This HA-mediated CTC clustering enables TNBC CTCs to overcome key obstacles, such as shear stress in the bloodstream, to ultimately enhance their survival and metastasis. Accordingly, depletion of HA by knockdown of its major synthase HAS2 suppresses CTC clustering and metastasis in mice.

While dysregulation of HA has been linked to multiple diseases, including cancer, these studies have primarily focused on HA-associated signaling pathways, such as HA-induced Akt activation through a positive feedback loop that promotes tumor cell survival[62]. However, the precise mechanisms by which HA contributes to the metastatic process are unclear. In particular, HA's role in CTCs remains

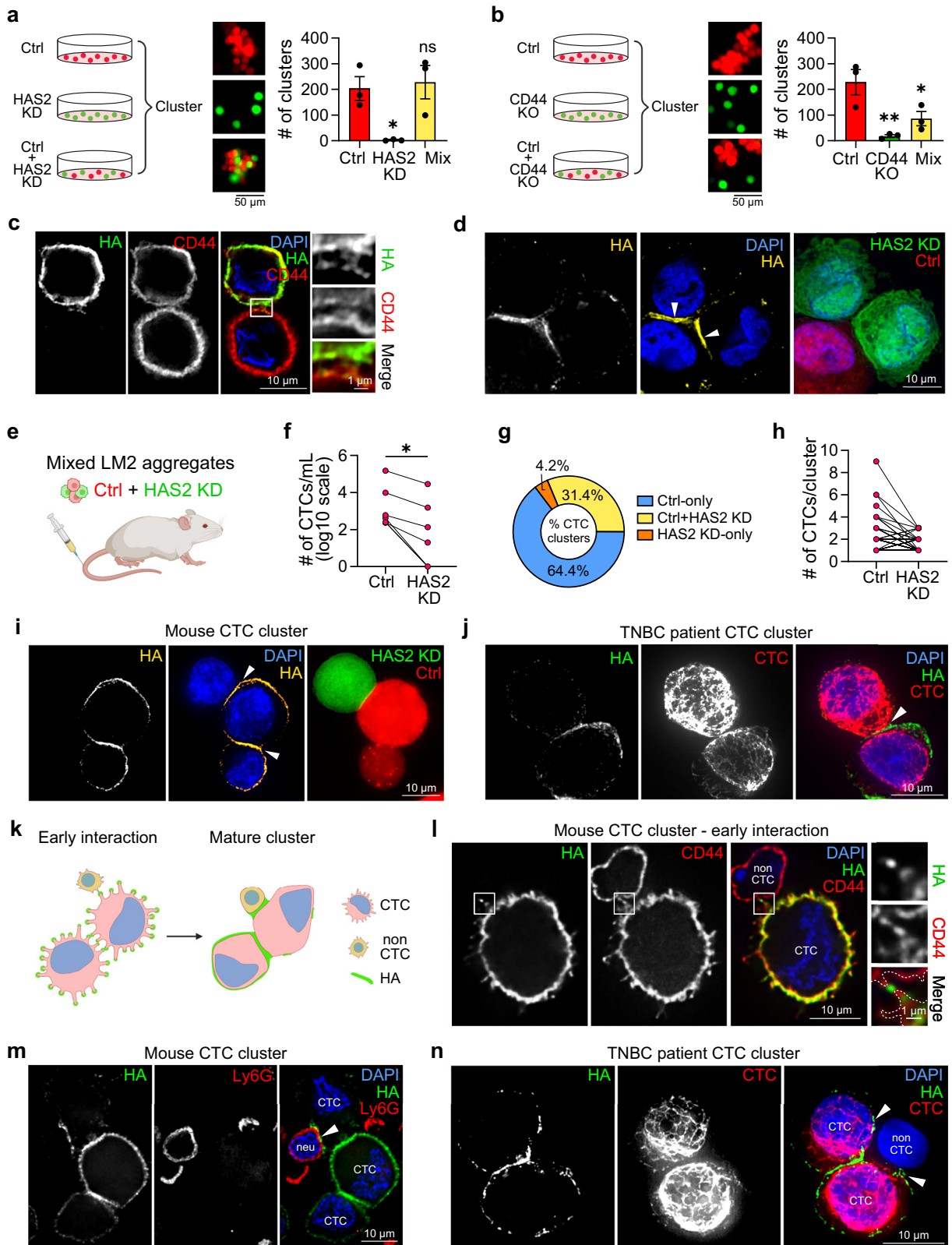

unexplored, which represents a significant gap in our understanding of HA's role in cancer progression. We show that beyond its established roles in promoting cell migration, cell proliferation[63], and pro-survival signaling[62,64,65], HA is essential for the stable clustering of TNBC CTCs in the absence of AJs. These CTCs assemble into clusters by traversing the bloodstream encased in a self-produced, stable HA coat that facilitates interactions with neighboring cells. The formation of this coat is likely

a consequence of HA's unique mode of production. HA is synthesized at the plasma membrane, and while the growing HA chain extrudes out into the extracellular space, it is immediately captured and bound by its receptor CD44[32]. In contrast, most ECM components are synthesized intracellularly, processed through the secretory pathway, and released as fully formed molecules[66]. While the ECM has traditionally been studied in its role as a structural network produced by and

**Fig. 6 | HA facilitates interactions between CTCs and non-CTCs. a** Left: schematic of LM2 Ctrl-alone, HAS2 KD-alone, or a 1:1 mixture of Ctrl (red) and HAS2 KD (green) single cells set up for clustering assay. Right: representative images and quantification of cluster numbers after 1 h of clustering (*n* = 3 biological replicates; *P* = 0.0225, 0.9958). **b** Left: schematic of LM2 Ctrl-alone, CD44 KO-alone, or 1:1 mixture of Ctrl (red) and CD44 KO (green) single cells set up for clustering assay. Right: representative images and quantification of cluster numbers after 1 h of clustering (*n* = 3 biological replicates; *P* = 0.0120, 0.0778) **c** Maximum intensity projection of fixed early-interacting LM2 control (top) and HAS2 KD (bottom) cells stained for HA (green pseudo color) and CD44 (red). Box indicates position of the 3-fold magnified region on the right demonstrating HA localization at interaction sites. See Supplementary Fig. 8c for staining of the HAS2 KD cell. **d** Maximum intensity projection of an in vitro LM2 cell cluster containing GFP-tagged HAS2 KD cells (green) and RFP-tagged Ctrl cells (red) stained for HA (yellow). White arrowheads indicate HA enrichment at the cell-cell interaction site between control and HAS2 KD cells. See Supplementary Fig. 8d for HA quantification. **e** Schematic depicting tail vein injection of LM2 cell aggregates in immunocompromised mice. Mixed aggregates of RFP-labeled Ctrl and GFP-labeled HAS2 KD cells were injected. **f** Quantification of the total number of CTCs originating from Ctrl and HAS2 KD cells per mouse. Each line represents one mouse (*n* = 7 mice; *P* = 0.0156). Two mice had overlapping values. **g** Pie chart showing average proportions of CTC clusters composed of only Ctrl cells, only HAS2 KD cells, or a mixture of both Ctrl and HAS2 KD cells (*n* = 3 mice). Mice with no CTC clusters were excluded from this analysis. **h** Quantification of the number of CTCs originating from Ctrl and HAS2 KD cells in each mixed CTC cluster. Each dot represents the number of Ctrl or HAS2 KD CTCs found in a mixed CTC cluster, with the connecting line indicating paired measurements (*n* = 108 total clusters from 3 mice; many clusters had identical values).

**i** Maximum intensity projection of a fixed LM2 CTC cluster containing RFP-tagged Ctrl cells (red) and GFP-tagged HAS2 KD cells (green) stained for HA (yellow). White arrowhead indicates HA enrichment at the cell-cell interaction site between control and HAS2 KD CTCs. **j** Maximum intensity projection of a TNBC patient CTC cluster consisting of two CTCs with heterogeneous HA (green) and CTC marker (red) staining. White arrowhead indicates HA enrichment at CTC-CTC interaction site. **k** Schematic of HA localization in early interactions and mature clusters between CTCs and non-CTCs. **l** Maximum intensity projection of a freshly established heterotypic LM2 CTC and non-CTC cluster stained for HA (green) and CD44 (red pseudo color). Box indicates the position of the 3-fold magnified region on the right which illustrates the HA-mediated connection of protrusions (outlined with dotted white lines) from adjacent cells. See Supplementary Fig. 8f for CTC staining. **m** Maximum intensity projection of a fixed heterotypic LM2 CTC cluster stained for HA (green) and Ly6G (red pseudo color). "neu" denotes Ly6G+ neutrophil. White arrowhead indicates HA enrichment at the CTC-neutrophil interaction site. See Supplementary Fig. 8h for CTC staining. **n** Maximum intensity projection of a patient CTC cluster consisting of two CTCs and a non-tumor cell. Cells were stained for HA (green) and CTC markers (red). White arrowheads indicate HA enrichment at CTC and non-CTC interaction site. Data are represented as mean ± SEM. Statistical significance: ns = not significant; *$P$ = < 0.05; **$P$ = < 0.01; ****$P$ = < 0.0001 (ordinary one-way ANOVA (**a**, **b**) or paired two-sided Wilcoxon test (**f**)). Blood from three different mice and five different patients was analyzed to obtain the representative microscopy pictures shown in **i**–**j**, **l**–**n**. DAPI (blue) served as a nuclear counterstain. Note: for visualization purposes, green/red pseudo-color were assigned for HA/CD44/Ly6G to match the color scheme used throughout the manuscript. **e**, **k** were created in BioRender. Bobkov, G. (2026) https://BioRender.com/srzuwrk.

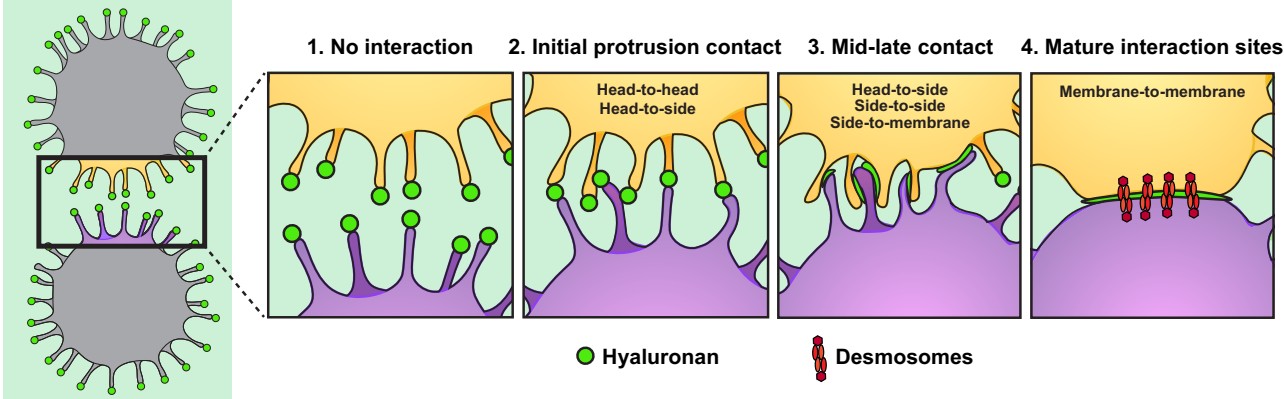

**Fig. 7 | Model depicting the HA-mediated cell clustering mechanism.** Two neighboring cells are shown on the left with blow-up illustrations on the right depicting the maturation process of an HA-based interaction site. Metastatic TNBC tumor cells form actin-based membrane protrusions with HA (green) localized at their tips (1). Neighboring cells initiate HA-mediated contacts at these protrusions (2), which subsequently glide alongside each other to form stable cell clusters (3, 4). HA stabilizes cell connections throughout this process. Subsequent assembly of desmosomes connects the interaction to intermediate filaments and mechanically stabilizes HA-based CTC clusters (4). Figure created by SciComm Consulting, LLC.

supporting tissue-resident cells, our study reveals that tumor cells in suspension in vitro and during circulation in vivo can retain their own ECM complement.

Highly aggressive TNBCs often exhibit mesenchymal features and downregulate epithelial AJ molecules to increase their invasive and migratory properties[15–18]. While this promotes dissemination, the lack of classical AJ-mediated interactions can prevent the formation of CTC clusters[67]. HAS2 has been previously linked to EMT phenotypic changes[62,68,69]. Our discovery of an HA-mediated clustering mechanism in TNBC CTCs highlights the adaptability of tumor cells and reveals how they can leverage both mesenchymal adaptations and CTC cluster formation to optimize their metastatic potential. A close examination of the HA-mediated and AJ-mediated cell interaction mechanisms reveals several parallels as well as distinctions. Although the

specific molecular components differ, the underlying principles of stable cell clustering observed in epithelial AJs[47] and HA-mediated cell-cell interactions are surprisingly similar.

For instance, the formation of HA-mediated interactions relies on actin-based protrusions, similar to the involvement of actin-based protrusions in the formation of AJs[46,47,70]. Moreover, in singular or early-interacting tumor cells, HA is enriched at the tips of these protrusions (Fig. 7, panels 1–2), which strikingly resembles the localization of the key AJ component E-cadherin during the formation of AJs[46]. As cells connect to each other through these protrusions, AJs expand the membrane area involved in cell-cell interaction, forming adhesion zippers[46]. Similarly, HA-based clustering is initiated by cellular protrusions, which slide alongside each other to gradually increase the membrane-membrane contact area (Fig. 7, panels 3–4). Given the repetitive composition of HA[71] and the abundant presence of CD44 at

the cell surface, stable HA-based cell-cell contacts are likely facilitated through numerous receptor-ligand interactions, especially as the binding affinity of a single CD44 to its ligand HA is relatively low[72,73]. The similarities further extend to the formation of desmosomes, which occurs only after initial membrane contacts have been established through AJs or HA. These similarities ultimately result in comparable structural features such as gap sizes between interacting cells and the stable connection of cells across a large membrane area[39,40].

While these cell interaction mechanisms share many overlapping features, they differ substantially in the flexibility of their cell-cell connections. Epithelial AJs provide a very strong, rigid connection that does not break easily. Accordingly, AJ-mediated clusters shed from primary tumors as intact clusters and are rapidly cleared from circulation[9], likely because they are trapped in narrow blood vessels. In contrast, HA-based CTC clusters are initiated through numerous weak receptor-ligand interactions[72,73] and can thus be more flexible. Their ability to form cell contacts within a short time period, even under shear stress, and to organize in a myriad of diverse structures from nearly round spheres to almost linear stretches indicate their dynamic nature. HA-mediated CTC clusters from more mesenchymal tumors likely form through shedding of intact clusters or through clustering of single cells prior to intravasation into blood vessels[11]. In addition, clusters may form while CTCs are in close proximity in circulation, either when trapped together in narrow blood vessels or upon exiting a narrow constriction. Indeed, elegant experiments using microfluidic devices and live-cell imaging in zebrafish and Drosophila demonstrated that some CTC clusters can pass through narrow structures similar in size to human capillaries[74–76]. Successful transit of these clusters involved dynamic reorganization of cell-cell interactions, allowing cells to pass through in a single file, followed by reformation of the cluster upon exit from the capillary-like structure[74–76]. As these results were obtained using the parental line of LM2 cells, MDA-MB-231[74], which also utilizes the HA-based clustering mechanism, it is highly likely that the dynamic reorganizations of HA-mediated cell-cell interactions enables CTC clusters to pass through narrow vessels and seed distant metastases. Importantly, HA-mediated and AJ-dependent cell-cell interaction pathways function independently, and the loss of HA-based clustering in HAS2 KD cells does not trigger compensatory upregulation of AJ components to restore clustering via an alternative cell-cell interaction mechanism.

Cancer metastasis is a highly complex process requiring tumor cells to overcome numerous stressors. While single CTCs are less likely to complete this process, cooperation among multiple cells in a CTC cluster has been shown to enhance the metastatic potential of clustered CTCs[77]. One of the most intriguing aspects of the HA-based CTC clustering mechanism is the seamless integration of diverse cell types into HA-mediated clusters. In principle, HA-mediated cell contacts can incorporate any cells expressing an HA receptor into the CTC clusters. Since CD44 is ubiquitously expressed and often upregulated during cancer progression[78], this mechanism can enable the inclusion of both heterogenous CTCs as well as non-CTCs into HA-based CTC clusters. Heterotypic CTC clusters containing white blood cells, such as neutrophils[3] and myeloid-derived suppressor cells[79,80], or cancer-associated fibroblasts[81], have been reported, and all of these cell types express CD44[22–24]. Moreover, this mechanism could also be the basis for the integration of acellular components such as CD44-expressing platelets[26] into CTC clusters. Indeed, we observed enrichment of CTC-derived HA at interaction sites between CTCs and non-CTCs in metastatic TNBC patients and mouse models. The inclusion of diverse cell types into CTC clusters can enhance their metastatic potential by promoting CTC proliferation, survival, immune evasion, and the preparation of the pre-metastatic niche[3,6,79,80,82,83]. Notably, not all CD44-expressing cells are integrated into heterotypic CTC clusters, and the precise regulation of this process remains to be determined in future investigations.

In conclusion, our work has uncovered a cell-cell adhesion mechanism that mediates stable cell contacts in a cell-ECM-cell manner. This mechanism is employed by TNBC CTCs that lack epithelial adhesion proteins and enables them to combine their migratory and invasive advantages with the enhanced metastatic potential conferred by CTC clusters. This dual advantage renders TNBC CTCs highly aggressive and highlights new possibilities for developing clinical interventions to treat metastatic TNBC patients.

## Methods

### Ethics
All animal procedures were performed with approval from the Institutional Animal Care and Use Committee at Baylor College of Medicine (Protocol number AN-7145). For human sample analysis, de-identified blood samples from breast cancer patients were obtained after receiving their informed consent as per the institutional review board (IRB) at Baylor College of Medicine.

### Cell lines and transfection
All cell lines were maintained at 37 °C in 5% $CO_2$ and passaged every 2–3 days. Base medium for all cell lines was supplemented with penicillin and streptomycin. MDA-MB-231, LM2, MCF7, 293FT, PANC-1, and DAOY cells were cultured in DMEM supplemented with 10% FBS. BT474 cells were cultured in RPMI 1640 supplemented with 10% FBS. SKBR3 cells were cultured in McCoy's 5 A supplemented with 10% FBS. BT549 cells were cultured in RPMI 1640 supplemented with 10% FBS and 10 μg/mL insulin. SUM159 cells were cultured in F12 supplemented with 5% FBS, 10 mM HEPES, 5 μg/mL insulin, and 1 μg/mL hydrocortisone (HC). MCF10A were cultured in DMEM supplemented with 5% horse serum, 20 ng/mL EGF, 0.5 μg/mL HC, 100 ng/mL cholera toxin, and 10 μg/mL insulin. WHIM12 cells were cultured in DMEM/F12 (1:1) supplemented with 10% FBS, 0.1X ITS-X (Thermo Fisher 51500056), 10 ng/mL EGF, 50 ng/mL 3,3',5-triiodo-L-thyronine, and 10 ng/mL HC. PC-3 cells were cultured in RPMI 1640 supplemented with 10% FBS, L-glutamine, and 10 mM HEPES. MDA-MB-231 and its derivative LM2 were obtained from Dr. Yibin Kang. HMLE from Dr. Jing Yang, MCF10A from Dr. Alexander Minella, BT474 from Dr. Charles V. Clevenger, SUM159 from Dr. Robert Weinberg, WHIM12 from Dr. Matthew Ellis, MCF7 from Dr. Marcus Peter, SKBR3 from Dr. Vince Cryns, PC-3 from Dr. Raymond Bergan, and DAOY from Dr. Richard Hurwitz. 293FT and BT549 were purchased from ATCC. PANC-1 was purchased from BCM Molecular and Cellular Biology Tissue Culture Core Laboratory.

For HA depletion experiments, cells were treated with 500 U/mL of crude Hyaluronidase from bovine testes (Sigma H4272) or 1.5 U/mL of purified Hyaluronidase from *Streptomyces hyalurolyticus* (Sigma H1136). For exogenous HA treatment, cells were pre-incubated for 1 h with the indicated concentration and size of HA (Sigma 75043 and 53163) in a low-attachment plate prior to clustering. To inhibit cytoskeletal dynamics, attached cells were treated for 1.5 h with 5 μM of Latrunculin A (Sigma 428026) or with 10 μM of Colchicine (Sigma C9754).

For transient expression, 293FT cells were transfected using the X-tremeGENE™ HP DNA Transfection Reagent (Roche 6366236001) and analyzed after 48 h. Retroviral or lentiviral infections were performed for generation of stable cell lines. Plasmids used for HAS2 expression (pLX304-HAS2) or shRNA-mediated HAS2 knockdown #1 (pLKO.1-shHAS2#1) were described previously[84]. The plasmid for shRNA-mediated HAS2 knockdown #2 (targeting sequence: 5′-GCCATGCTT-TATGTGGGTTAT-3′) was cloned into pLKO.1 backbone using AgeI and EcoRI restriction enzymes. shHAS2 #1 was used for experiments unless otherwise specified. The plasmids for DSG2 knockdown (targeting sequence: 5′-GCTCAAACTAACGAAGGAATT-3′), DSC2 knockdown (targeting sequence: 5′-TAGCAGTGGCATAAGGTATAA-3′), and DSP knockdown (targeting sequence: 5′-GCAGAATGATTCACAAGCAAT-3′)

were cloned in the same manner. EFS-GFP was a gift from Michael Lewis (Addgene plasmid #110834)[85]. pWPT-RFP was kindly provided by Xiang Zhang. These two plasmids were used to tag LM2 cells with GFP or RFP. pBRIT-CD44-HA was used to express CD44 in 293FT cells[86]. The R41 residue of CD44 was mutated to Alanine using the Quick Protocol for Q5® Site-Directed Mutagenesis Kit (NEB E0554) according to the manufacturer's instructions. Lentiviral CRISPR/Cas9 was used to generate LM2 CD44 KO cells[87] with a gRNA targeting exon 1 of CD44 (fwd:5'-CACCGAGTTTTGGTGGCACGCAGCC-3'; rev:5'-AAACGGCTGC GTGCCACCAAAACTC-3').

## Antibodies and other reagents

For immunofluorescence, the following antibodies were used: CD44-IM7 (Santa Cruz Biotechnology sc-18849; 1:100), DSG2 (Proteintech 21880; 1:100), DSC2 (ABclonal A10211; 1:100), DSP (ABclonal A7635; 1:100), Tubulin (Proteintech 11224-1-AP; 1:100), Ki67 (eBioscience 14-5698-82; 1:100), GFP (Abcam ab13970; 1:100), RFP (Rockland 600-401-379; 1:100), Ly6G-AF594 (Biolegend 127636; 1:50), Pan-cytokeratin (Santa Cruz Biotechnology sc-8018; 1:100), Cytokeratin 19 (Invitrogen MA5-12663; 1:100), EpCAM-AF488 (Cell Signaling 5198S; 1:50), and EGFR-AF488 (Biolegend 352908; 1:50). Biotinylated HA binding protein (Millipore 385911; 2 μg/mL) was used to detect HA and AF568-coupled phalloidin (Thermo Fisher A12380; 1:100) to detect actin. For immunoblotting, the following antibodies were used: CD44 (R&D BBA10; 1:1000), HA tag (Cell Signaling 3724S; 1:1000), E-cadherin (Cell Signaling 3195S; 1:2000), HAS2 (Santa Cruz Biotechnology sc-514737; 1:500), DSG2 (Proteintech 21880; 1:1000), DSC2 (ABclonal A10211; 1:1000), DSP (ABclonal A7635; 1:1000), GAPDH (Millipore MAB374; 1:3000), and β-actin (Sigma A1978; 1:3000).

## Bioinformatics analysis

The TCGA BRCA RNAseq expression dataset was downloaded from the GDC portal (https://portal.gdc.cancer.gov). METABRIC breast cancer expression dataset[88,89] was downloaded from the cBioPortal (https://www.cbioportal.org). CCLE expression dataset was downloaded from Broad Institute's DepMap Portal (https://depmap.org/portal). Each dataset's internal PAM50 classification was used to divide tumor samples by subtype. For the heatmaps, z-score normalization was performed for each gene across all samples and plotted. Differential gene expression analysis of TNBC vs. non-TNBC tumors was conducted using DESeq2 performed on genes with at least 5 counts present in at least half of the samples. To determine the differential gene expression of tumors expressing low vs. high levels of E-cadherin, the bottom 10% (E-cadherin_low) and top 10% (E-cadherin_high) of all breast cancer samples based on their E-cadherin expression were compared. To characterize the cancer processes associated with HAS2, we first filtered out low abundance genes that have less than 5 reads across half of the samples in TCGA BRCA. We then applied log2-tranformed counts per million (CPM) normalization and the trimmed mean of M-values (TMM) adjustment on the count matrix, after which correlation coefficient of HAS2 expression with each gene was calculated using Pearson correlation. GSEA was performed with our in-house script, using fgsea on genes ranked by log2FC with 1000 gene set level permutations and considering gene sets of size 15–500 genes. Gene ontology (GO) cellular component analysis and GSEA KEGG pathway analysis were both conducted using WebGestalt (https://www.webgestalt.org) with the default settings[90]. For GO analysis, the top 500 upregulated genes were uploaded. For GSEA, the correlation coefficient of the whole transcriptome to HAS2 gene expression was used to pre-rank the genes. We utilized our own curated signatures as well as metastasis signature derived from the C2 collection in MSigDB (https://www.gsea-msigdb.org/gsea/msigdb).

## Cluster assay

Cells were detached using TrypLE Express (Gibco 12605010) and resuspended in fresh growth media to obtain a single cell suspension. 100 K cells were transferred to 500 μL of growth media in a 12 well plate, which was immediately placed on an orbital shaker at 37 °C in 5% $CO_2$. Alternatively, 50 K cells were placed in 250 μL of growth media in a 24 well plate, keeping the ratio of number of cells to total volume consistent. The plate was agitated at 200 rpm for 1 h. Afterwards, each entire well was scanned using the IncuCyte® S3 live imaging system (Sartorius). ImageJ (v1.54.f) was used to quantify the number and size of single cells and clusters. Particles between 1 and 100 pixels² were categorized as single cells, while particles greater than 700 pixels² (roughly representing a cluster of 5–10 or more cells) were categorized as clusters.

For no shear stress condition, 50 K cells were seeded in 250 μL of growth media in a low-attachment 24 well plate and allowed to aggregate for 24 h. Plates were scanned using IncuCyte as described above to quantify no shear stress, or 0 rpm, clustering. Next, these plates were placed on an orbital shaker and agitated at increasing speeds in 50 rpm increments, from 100 rpm to 350 rpm. For desmosome knockdowns, stability of clustering cells was examined by first inducing cluster formation for 30 min at 200 rpm, followed by agitation at increasing speeds in 50 rpm increments from 250 rpm to 350 rpm. For each speed, plates were agitated for 15 min, imaged using IncuCyte to measure changes in clustering, and subsequently placed back on the shaker at the next speed. Alternatively, after 6 h of aggregation in low-attachment plates and imaging, cells were agitated at 200 rpm for 1 h and imaged.

## HA-CD44 interaction blocking assay

Cells were seeded in a 6 well plate and allowed to grow to 70% confluency for two days. At this point, the old media was replaced with 1 mL of fresh growth media containing 10 μg/mL of Rat IgG control (BioXcell BE0089) or CD44-IM7 (BioXcell BE0039) antibody. After 3 h of antibody treatment, cells were detached to perform the cluster assay.

## Quantitative RT-PCR

RNA was extracted using the E.Z.N.A Total RNA Kit I (Omega Bio-Tek), and cDNA was generated using GoScript Reverse Transcription System (Promega). qRT-PCR was performed using GoTaq qPCR Master Mix (Promega). For every biological replicate, mRNA expression of two technical replicates was averaged and normalized to mRNA expression of TATA-binding protein (TBP). The following primers were used: HAS2 (fwd:5'-AGAGCACTGGGACGAAGTGT-3'; rev:5'-ATGCACTGAA-CACACCCAAA-3'), CD44 (fwd:5'-TACTGATGATGACGTGAGCA-3'; rev:5'-GAATGTGTCTTGGTCTCTGGT-3') and TBP (fwd:5'-GGA-GAGTTCTGGGATTGTAC7-3'; rev:5'-CTTATCCTCATGATTACCG-CAG-3').

## Immunoblotting

Cells were harvested in RIPA lysis buffer (2% NP-40, 1% Sodium Deoxycholate, 0.2% SDS supplemented with 1x protease inhibitor cocktail in PBS) and lysed for 30 min on ice. Lysates were cleared via centrifugation, and protein concentration was measured using Bradford protein assay (Bio-Rad 500-0006). For the HAS2 blot, instead of loading equal amounts of protein, lysates derived from the same amount of cells were not cleared through centrifugation but directly boiled in SDS sample buffer and loaded in equal volumes. Samples were boiled in SDS sample buffer, and 20 μg of protein per sample was subjected to electrophoresis by 4–20% SDS-PAGE (Genscript) and transferred to a methanol activated PVDF membrane (GE 10600023). Membranes were blocked with 5% milk in TBST for 30 min at room temperature (RT) and then incubated in primary antibodies overnight at 4 °C. The next day, membranes were washed thrice in TBST and incubated in the

corresponding HRP-conjugated secondary antibodies for 1 h at RT. After final TBST washes, membranes were imaged using a ChemiDoc (Bio-Rad) with Immobilon Western Chemiluminescent HRP Substrate (Millipore).

## Immunofluorescence

Cells were either grown and fixed on glass coverslips or detached using TrypLE Express (Gibco 12605010) and fixed directly in solution as single cells or subjected to the clustering assay for 15–90 min. For membrane staining, 1000x dye stock solution of CellBrite® Fix 555 Membrane Stain (Biotium 30088) was added at a 1:400 dilution for the last 15 min of the clustering assay. Tumor cell clusters were transferred into a µ Slide 8 well Glass Bottom (Ibidi 80827), and after 5 min of settling onto the slide, liquid was carefully removed while observing the chamber under a light microscope. This results in spreading of the clusters across the glass bottom when surrounding liquid disappears. Before fixation, remaining liquid was allowed to evaporate for 5–10 min.

For all staining, fixation was performed for 10 min at RT in PBS containing 4% Paraformaldehyde (Fisher Scientific AA433689M) and 0.1% Glutaraldehyde (Electron Microscopy Sciences 16019). When required, samples were permeabilized for 10 min in PBS/0.1% Triton and endogenous biotin was masked using Avidin/Biotin Blocking Solution (Sakura #10-0039). Cells were blocked for at least 1 h in PBS/ 2% Bovine Serum Albumin (VWR 0332-25 G) and incubated in primary antibody and biotinylated HA binding protein (Millipore 385911) overnight at 4 °C. Following three washes in PBS, samples were incubated in PBS containing the respective Alexa Fluor (AF)-coupled secondary antibodies and/or Streptavidin AF647 (Molecular Probes S21374) for 1 h at RT. After three additional washes in PBS, samples were mounted in ProLong™ Diamond Antifade Mountant with DAPI (Thermo Fisher P36971). For staining with any primary AF-conjugated antibodies, samples were incubated at RT for 2 h. Note: to keep the structure of IF images consistent throughout the manuscript, green pseudo color was chosen for HA, despite it being imaged in the far-red channel using Alexa Fluor 647-coupled streptavidin. For presentation purposes, pseudo green and red colors were also used in samples that included GFP-/RFP-positive cells.

## Microscopy and image analysis

Images were acquired using a GE Healthcare DeltaVision LIVE High Resolution Deconvolution Microscope and were deconvolved using softWoRx Explorer or an Olympus IX83 epifluorescence deconvolution microscope. Images of fixed cells were taken as 50–65z stacks of 0.2 µM increments using a 100× oil immersion objective. Quantification of signal intensities was performed using Fiji[91]. Enrichment of HA/ CD44 was quantified by averaging signal intensities of five different areas across the cell-cell interaction site ("cell junction") or five different areas that are not in contact with neighboring cells ("non-junction"). For enrichment of HA/total Membrane, an equal number of "cell junction" and "non-junction" regions were quantified for HA and membrane signal, and HA signal divided by membrane signal. Protrusion length was determined by averaging the lengths of the five largest protrusions per cell.

Time-lapse imaging was performed with 25z stacks of 0.4 µM increments using a 60× oil immersion objective and a time-lapse of 5 min. Cells were detached as described for the cluster assay and transported as a single cell solution to the microscope. There, cells were transferred to a µ Slide 8 well Glass Bottom (Ibidi 80827) and the imaging run was initiated.

## Electron microscopy

Cells were detached as described for the cluster assay and either placed in 6-well low-attachment plates and treated with DMSO or HAse for 16 h to induce gravity-based cell aggregation or clustered in the

cluster assay for 5, 10, and 20 min to capture early-to-mid clustering events. Formed clusters were transferred into tubes and spun down at 500 g for 5 min. Pelleted clusters were placed into scintillator vials with 2.5% Paraformaldehyde and 2% Glutaraldehyde in 0.1 M Sodium Cacodylate buffer and placed on a rotator in a cold room overnight. The next day the cell pellets were processed inside a Ted Pella Bio Wave Vacuum Microwave and cells were re-pelleted after initial reagent changes. Samples were fixed again for 1 min at 650 watts, followed by 3 rinses with 0.1 M Sodium Cacodylate buffer for 40 seconds at 150 watts. Pellets were post-fixed at 100 watts for two 2-min on-off-on cycles of 1% buffered Osmium Tetroxide. Pellets were then rinsed thrice with Millipore water at 150 watts for 40 s. Aqueous Ethanol concentrations from 25 to 100% were used as the initial dehydration series for 40 s each at 150 watts, followed with 3 Propylene Oxide changes as a final dehydrant for 40 s at 150 watts each. Samples were gradually infiltrated using a graded series of Propylene Oxide and Embed 812 resin for 3 min each at 250 watts. Finally, the samples were infiltrated with 3 changes of pure resin at 250 watts for 3 min each under vacuum. Samples were allowed to infiltrate further in pure resin overnight on a rotator at RT. The samples were embedded into regular Beem capsules and cured in the oven at 62 °C for 3 days. The polymerized samples were sectioned into 50–55 nm silver sections. Blocks were thin sectioned at 50–55 nm silver sections on a Leica UC7 Ultra-microtome and placed on formvar coated copper slot grids. Sections were stained with 1% Uranyl Acetate for 13 min and Lead Citrate for 2 min. Micrographs were taken on a JEOL 1400Plus TEM with a mid-mount AMT R16 TEM camera.

## Proliferation assay

25 K LM2 control and HAS2 KD cells were seeded per well in 24 well plates in normal growth media. Plates were scanned using IncuCyte® S3 live imaging system every 4 h for 2 days. Cell confluency was analyzed at each time point to generate a cell proliferation curve.

## Cell survival assay

Cells were detached using TrypLE Express and resuspended in DMEM supplemented with 0.5% FBS. 100 K cells in 500 µl of DMEM supplemented with 0.5% FBS and either DMSO or HAse were seeded in each well of a 24 well plate, which was immediately placed on an orbital shaker at 37 °C in 5% $CO_2$. Cells were agitated at 200 rpm for 6, 16, or 24 h. For each condition, cells from 3 wells were combined to obtain enough material for analysis. For the rescue experiment, cells were treated with 5 mM N-acetylcysteine (NAC; Sigma A9165) during the 24 h of agitation. At the end of the timepoint, cells were first manually counted with a Hemocytometer to determine the remaining number of cells in each sample. Then, they were spun down in flow tubes at 500 g for 5 min. Pellets were resuspended into a single cell suspension in PBS and stained with Propidium Iodide (PI; BD Pharmingen 556463, 1:250 dilution) for 15 min at RT in the dark. Samples were immediately run on CytoFLEX benchtop flow cytometer (Beckman Coulter CytExpert v2.4.0.28) to assess PI staining and FlowJo v10 was used to quantify cell viability.

For ROS measurement, cells were prepared similarly, but they were agitated for 12 h. At the same time, a portion of the detached cells ("wild type") was placed in a low-attachment plate without any agitation to use for signal normalization. At the end of the timepoint, cells were incubated in PBS containing ROS Deep Red dye (Abcam ab186029; 1:1000) for 30 min at 37 °C in the dark. CytoFLEX benchtop flow cytometer and FlowJo were used to determine ROS levels.

## Matrigel invasion assay

LM2 cells were starved in serum-free DMEM for 24 h prior to this assay. To generate tumor cell clusters, 25 K of the starved LM2 control cells were placed in a low-attachment 24 well plate in serum-free DMEM and subjected to the clustering assay for 1 h at 200 rpm. For the HAse

condition, 25 K of the starved LM2 cells were incubated in suspension with HAse in serum-free DMEM for 1 h at 37 °C, without any shear force. Next, control clusters and HAse-treated single cells were transferred to Matrigel invasion chambers (Corning 354480), which had been equilibrated for 2–4 h at 37 °C in serum-free DMEM. At the same time, DMEM supplemented with 10% FBS was added as a chemoattractant to the wells of the 24-well plate containing the Matrigel chambers. Cells were allowed to invade through the Matrigel to the exterior of the chamber. After 24 h, cells from the interior of the chamber were removed using a cotton swab, while cells on the exterior were stained using crystal violet dye. Invading cells were imaged using a light microscope and quantified via ImageJ (v1.54.f).

## Mice

Mice were housed in a 14:10 h light/dark cycle at an ambient temperature of 20–23 °C. Humidity was maintained between 30-70%. NSG mice (NOD.Cg-Prkdcscid Il2rgtm1Wjl/SzJ) were originally obtained from Jackson Laboratory. Female mice were utilized in this study as breast cancer primarily occurs in females. For spontaneous metastasis experiments, 10 K LM2 control and HAS2 KD cells were injected with Matrigel (Corning 354230) into the fourth mammary fat pad of 6–9 weeks old NSG female mice. Primary tumors were monitored twice a week and measured using a caliper. Tumor volumes were calculated as half of length × width$^2$. The maximum permitted tumor size of 1.5 cm was not exceeded. After 58 days, mice were euthanized, and blood was collected via heart puncture into EDTA tubes. Primary tumors were removed and weighed individually. Additionally, lungs were collected after perfusion with 10% neutral buffered formalin (NBF) and fixed for H&E or immunofluorescent staining. To obtain WHIM12 PDX primary tumor cells for in vitro studies, 6–8 weeks old NSG female mice were implanted with a 1–2 mm$^2$ tumor chunk (only passaged in mice) in the fourth mammary fat pad. Tumors were collected prior to reaching 800 mm$^2$ and dissociated into single cells.

For tail vein experiments, 100 K LM2 control and HAS2 KD cells were injected as aggregates into 6–8 weeks old NSG female mice by tail vein. Within 20 min of the injection, baseline cell amount in the lungs was quantified using Luciferin (Gold Biotechnology LUCK-100) injection and IVIS spectrum imaging (Caliper LifeScience). Lung metastasis BLI signal was monitored every week. After 4 weeks, mice were euthanized, and blood and lungs were collected and processed as described above. For the short-term 24 h tail vein experiment, $1 \times 10^6$ LM2 control and HAS2 KD cells were injected as aggregates, and lungs were collected and processed as described above. ImageJ (v1.54.f) was used to quantify the percentages of tumor cells in lungs. For each mouse, average tumor cell percentages from 12 random locations are reported.

For CTC analysis, whole blood was centrifuged at $700 \times g$ for 15 min to deplete plasma. Pellets were resuspended in ACK lysing buffer (Thermo Fisher A1049201) and incubated for 3–5 min to lyse red blood cells. Samples were diluted with PBS and centrifuged at $700 \times g$ for 5 min to obtain pellets containing white blood cells and CTCs. Pellets were resuspended in PBS, spread onto slides coated with Poly-L-Lysine (Electron Microscopy Sciences 19320-A), and allowed to attach for 30-60 min at 37 °C. Slides were fixed and imaged using a Cyte-Finder® instrument (RareCyte) to quantify GFP- or RFP-labeled CTCs. For each mouse, CTC numbers quantified from a portion of the pellet were extrapolated to determine total CTCs per mL of blood.

## Patient CTC samples

Breast cancer primarily occurs in females, so human samples from female breast cancer patients were utilized in this study. For CTC analysis, blood was obtained from de-identified patients with metastatic TNBCs after receiving their informed consent as per institutional review board (IRB) protocol at Baylor College of Medicine. 6–8 mL of peripheral blood was collected into EDTA vacutainers. The blood was

centrifuged at $700 \times g$ for 15 min to deplete plasma. The pellet was resuspended in PBS, and peripheral blood mononuclear cells (PBMCs) were isolated using Lymphoprep™ density gradient medium (STEM-CELL Technologies 07851) and ACK lysing buffer. PBMCs were resuspended in Transfer Fluid (RareCyte) and spread onto slides. After fixation with 10% NBF for 20 min, immunofluorescent staining was performed as described above. CTCs were detected using a cocktail of four antibodies consisting of Pan-Cytokeratin, Cytokeratin 19, EpCAM, and EGFR. Slides were scanned using a CyteFinder® instrument (RareCyte) to identify CTCs.

## Statistics and reproducibility

GraphPad Prism was used to perform all statistical tests as described in the figure legends. $P$-value < 0.05 was considered statistically significant. $P = < 0.05$ (*), $P = < 0.01$ (**), $P = < 0.001$ (***), $P = < 0.0001$ (****) were indicated with exact values provided in the figure legends when possible. Additional statistical metrics, including confidence intervals, are provided in the Source Data file. Each experiment was replicated and successfully reproduced at least three times. All microscopy images and western blots represent results from these biological replicates. No statistical method was used to predetermine sample size for in vitro experiments. Mouse experiments were performed with groups of 7–8 mice based on pilot experiments and to account for any mortality prior to the completion of the experiment. No data was excluded except in the event of animal mortality prior to completion of the experiment. Mice and cell line samples were randomly allocated into groups to minimize bias. Investigators were blinded to group allocations during both data collection and data analysis for all mouse experiments and when possible, for in vitro experiments.

## Reporting summary

Further information on research design is available in the Nature Portfolio Reporting Summary linked to this article.

# Data availability

Source Data are provided with this paper and all the data that support the findings of this study are available within the Article and its Supplementary Information. Source data are provided with this paper.

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

## Acknowledgements

We thank the Cheng lab members for discussion and constructive feedback on this project. We thank the patients for supporting this study. We thank Alphi Kuriakose and Bryant McCue for coordinating and processing patient samples. We thank Shigenori Hirose and Mariko Katoh-Kurasawa for their assistance in developing the cell cluster assay. We thank Guillaume Jacquemet and Gautier Follain from University of Turku and Åbo Akademi University for project discussion. Biorender.com or SciComm Consulting, LLC was used for graphic creations. This research was supported by grants from NIH T32GM08307 and T32ES027801-06 (K.J.P), T32ES027801-05 and Frank & Sandra Kimmel Postdoc Endowment (G.B.), and R01CA276432 (C.C). C.C. is a Cancer Prevention

Research Institute of Texas Scholar in Cancer Research (RR160009). The project was assisted by the Integrated Microscopy Core (NIH DK56338, CA125123, ES030285, S10OD030414, S10OD020151; CPRIT RP150578, RP170719), the Jan & Dan Duncan NRI TEM Core (NIH P50HD103555), the RNA In Situ Hybridization Core facility (NIH 1S10OD016167), and the Cytometry and Cell Sorting Core (NIH CA125123, RR024574; CPRIT RP180672) at Baylor College of Medicine.

## Author contributions

G.B. and K.J.P. contributed equally to this work, and the authors' names were placed alphabetically. G.B., K.J.P., and C.C. designed experiments. G.B. and K.J.P. performed experiments. R.Z. assisted with bioinformatics analysis. B.M.L. assisted with the mouse experiments. M.J.E. provided the patient samples. G.B., K.J.P., and C.C. wrote the manuscript. G.S. provided advice to the project and edited the manuscript. C.C. supervised the project.

## Competing interests

M.J.E. reports income from Veracyte and Bioclassifier LLC and employment with AstraZeneca unrelated to this work. All other authors declare no competing interests.
