## [Transparent Peer Review file · Nature Communications]

Extracellular matrix mediates circulating tumor cell clustering in triple-negative breast cancer metastasis

Corresponding Author: Dr Chonghui Cheng

Version 0:

Reviewer comments:

Reviewer #1

(Remarks to the Author)

Overall impression:

This well-written manuscript provides an interesting and useful extension of prior research documenting the importance of CD44 in TNBC metastasis by providing compelling evidence that the main ligand of this cell surface receptor, hyaluronan (HA), plays an important role in cluster formation. While the evidence for CD44 and HA in cluster formation is extensive and convincing, the importance of this interaction in promoting metastasis is less clear.

Major points:

1. The authors state that “While similar numbers of both control and HAS2 KD cells reached the lungs upon injection, clear differences in metastatic lung signal are detectable from one week onwards (Fig. 5k-l and Extended Data Fig. 5h).” At the same time, the authors show that both control and HAS2 KD cells exhibit the same level of proliferation and Ki67 staining of the metastatic lungs was the same between LM2 control and HAS2 KD tumor cells (Extended Data Fig. 5i). Why would there be a difference in lung metastasis over time if cells’ ability to colonize the lungs and their proliferation rate was the same? Week 1-4 typically represents outgrowth of cells that have successfully colonized the lungs. Shouldn’t there be a difference in ability to colonize the lungs given that HAs-treated cells exhibit a 2-fold increase in ROS levels compared to control cells? In fact, the authors’ in vitro results show that “HA depletion decreased cell survival, with significant differences being detected at 16 hrs. At 24 hrs, only 20-25% of the HAs-treated or HAS2 KD LM2 cells were alive, compared to 65% of the control cells (Fig. 4e-f).”

2. In Figure 6f, the authors analyzed the number of CTCs originating from control and HAS2 KD cells in each mouse and observed a 4- to 12-fold reduction of CTCs originating from HAS2 KD cells. Why was a 4-week time-point chosen? Again, one would expect there to be a significant decrease in CTCs within 24 h based on the author’s in vitro data and previously published work (see PMID: 31485072, which the authors failed to cite in this manuscript).

In addition, the authors need to rule out the possibility that even though HA is crucial for cluster formation, the final cell-cell contacts might be mediated by an additional cell-cell adhesion mechanism.

3. The authors state that “gap size is similar to the 10-20 nm reported for AJs; however, the cluster-forming metastatic cell lines lack the crucial AJ component E-cadherin.” Later, they also state that “LM2 control and HAs-treated cells showed similar numbers of clusters and single cells in the absence of shear stress (Fig. 4a and Extended Data Fig. 4a-c, “no shear stress”).” Although the evidence for the crucial involvement of HA in cluster formation is compelling, the authors have not ruled out the possibility that after HA-mediated cell attachment occurs, the final cell aggregates are held together most tightly by an alternative cell adhesive mechanism besides HA, even though E-cadherin is missing. Have the authors checked the expression level of other AJ components in LM2 cells? The data presented strongly suggests that another AJ component is involved in cell clustering. What happens if that component is knocked down? Can cells still form cell clusters effectively in response to shear stress?

Minor points:

1. An interesting observation is that freshly detached MCF7 cells fail to cluster when rotated on an orbital shaker. MCF7 cells express high levels of E-cadherin and form strong cell-cell contacts in the absence of shear force. If MCF7 cells are allowed to cluster and are then subjected to shear force, do they remain together as clusters? How strong is this interaction compared to HA-mediated cell clustering exhibited by LM2 cells? How do LM2 cells form clusters more readily in response to shear stress than MCF7 cells? Does HA act like a ‘glue’ making it more likely for colliding cells to interact with each

other? Do E-cadherin based clusters take a long time to form whereas HA-CD44 form more quickly?

2. Increased HA intensity at cell-cell junctions may be due to the presence of more plasma membrane. For example, CAAX-GFP and GPI-GFP intensity is typically greater at cell-cell junctions. A better measure of HA intensity at cell-cell junctions would be to normalize the signal by the total amount of plasma membrane (i.e., HA/plasma membrane staining).

3. Have the authors performed an outlier test for Extended Figure 3d? If the abnormally low data point for colchicine is removed, there is likely a significant increase in the number of single cells relative to control. Is the amount of CD44 on the plasma membrane altered when the microtubule network is dismantled? Perhaps the amount of HA on the plasma membrane is reduced?

4. Line 84: Has hyaluronan been identified previously among ECM components reported to be upregulated in CTCs?

5. Lines 236-237: The evidence presented does not show conclusively that “actin filaments provide the protrusive force for the formation of these structures” but instead that actin is necessary for their maintenance.

Wording:

1. Line 75: Typo: emboli rather than embolies

2. Line 232: The “remarkable overlap” is not surprising because protrusions are normally filled with actin, so actin should overlap with any cell surface or matrix component on the outside of protrusions.

3. Line 482: The wording appears too general: the non-tumor cells also need to be CD44-expressing.

Reviewer #2

(Remarks to the Author)

This manuscript by Bobkov et al, identified a previously unrecognized HA-mediated mechanism that enables stable CTC cluster formation in triple-negative breast cancer (TNBC). The study is well designed, integrating both in vitro and in vivo assays with human samples and mouse models. The data are convincing, and the findings are both novel and highly clinic relevant. The results expand the current paradigm of CTC cluster biology and may provide new possibilities for treating metastatic TNBC patients. Some comments are shown below to improve the manuscript:

Major comments:

1. In most of the images showing cell clustering (e.g., Fig. 1f, Fig. 2b, 2g, etc.), the non-clustered groups have less cells than the clustered groups. It would be better to show full images of the entire culture to demonstrate that each group started with a similar number of cells.

2. Please carefully double-check all image labels. For example, in Fig. 5k, the signal based on the bar shown should be between 150–2,000 on Day 0, but the y-axis is labeled as 10^4 in Fig.5i. In addition, the Week 4 raw signal appears to be between 2,000–20,000(Fig.5K), so it is unclear how the “signal (relative to Day 0)” can be ~1,000 (fig.5i).

3. The inclusion of patient-derived CTCs is a strength, but the sample size is not specified. In addition, including CTCs from other breast cancer subtypes for comparison, if feasible, would further enhance the translational relevance of the findings.

4. In addition to the RT-PCR data, the knockdown efficiency of HAS2 should be validated by Western blot. Since the pooled population contain residual HAS2, and the remaining HA can mediate cell clustering in HAS2 knockdown cells (as shown in Extended Data Fig. 5l). Providing this additional validation would strengthen the conclusions regarding HAS2's role.

5. In the conclusion, it is stated that: “The formation of this coat is likely a consequence of HA's unique mode of production. HA is synthesized at the plasma membrane, and while the growing HA chain extrudes out into the extracellular space, it is immediately captured and bound by its receptor CD44. This enables the formation of an HA-rich pericellular coat.” Could the authors provide citation to support this statement? Otherwise, additional experiments would be needed to validate it.

6. It is known that CTC clustering can promote cell survival, and HA has also been reported to support cell survival. Therefore, the cell survival data may be considered to remove or move to the supplemental data, as it is not central to the focus of the manuscript. In addition, the reactive oxygen species (ROS) data is weak. If these data are kept, the authors could strengthen this aspect by performing rescue experiments—for example, testing whether antioxidants can prevent the death of Hase-treated cells.

Minor comments:

1. Since SKBR3 cells are used in Fig. 1g, it would be more appropriate for the representative image in Fig. 1f to also show SKBR3 cells, rather than BT474 cells. This would help maintain consistency between the panels.

2. In the text, it is stated that “CTC clusters were detected in all 6 control mice”. However, only 5 mice are shown in fig.5h& 5i.

3. Since the tumor cells were labeled with GFP, it is unclear whether the GFP signal will be preserved during IF staining, especially given that HA is also detected using a green fluorophore. Could the authors clarify how these signals are distinguished in the images?

Reviewer #3

(Remarks to the Author)

In this manuscript, Bobkov and Patel et al. propose a novel role for hyaluronan in circulating tumor cell (CTC) clustering

during TNBC metastasis. They show that CTCs express HAS2, which synthesizes HA, and that HA interacts with CD44 on cancer cells to function as adherent junctions, which is very interesting and novel. Overall, the data are clear and well presented. However, some of the results sections are somewhat lengthy and would be better reorganizing or moving parts into the discussion. It is intriguing that HA appears to show such a strong effect as a mediator of cell-cell interactions, the data overall support this hypothesis well. Detailed comments are as below:

1. HA is a well-known serum marker for several cancers, including breast, prostate, and gastrointestinal cancers. Elevated serum HA levels have been correlated with a worse prognosis and increase risk of metastasis, which supports the findings of this study. However, this also indicates that HA is already abundant in the bloodstream of patients with cancer or tumor-bearing mice, suggesting that CTCs may not be the primary source of HA. Moreover, HA is highly accumulated in the primary tumor, raising the possibility that HA-coated cancer cells could detach from the primary tumor, intravasate, and form clusters. According to this study, endogenous HAS2 expression is crucial for HA-mediated CTC clustering (HAS2 overexpression and knockdown experiments). In the context of rich HA in the bloodstream or culture media, CTCs can form clusters independently of HAS2 expression? Can you also compare HAS2 expression or HA intensity in LM2 clusters compared to LM2 single cells?
2. Metastatic tumors often lose adherens junction expression as they undergo epithelial-to-mesenchymal transition. This transition is a very important process in metastasis. In this study, EpCAM-negative cancer cells show high HAS2 expression, whereas EpCAM-positive cancer cells exhibit little to no HAS2 expression (Fig. 1j and extended Fig.1l). Is there any evidence linking the HAS2 or HAS2-HA -CD44 axis to epithelial-to-mesenchymal phenotypic changes? And could this transition also be associated with CTC clustering or cell-cell interactions?
3. While this study primarily focused on the TNBC metastasis, it would be important to clarify whether similar observations are found in other aggressive or malignant cancer types.
4. HA exists in both high-molecular weight and low-molecular weight forms, which are known to play distinct roles in pathophysiology. The authors should determine the size of HA to better understand the novel function of HA.
5. The authors suggest that tumor cells express HAS, which synthesizes HA, and that this HA interacts with CD44 on cancer cells, forming a kind of autocrine loop. This axis mediates cell-cell interactions and promotes CTC clustering. The HA-CD44 interaction has been extensively studied in cancer biology. Do HA-coated CTC clusters also contribute to extravasation, cancer cell migration, or invasion, at later step of metastasis? In vivo data suggest that HA-mediated CTC clustering plays an important role in TNBC lung metastasis (Fig.5). Given that HA-CD44 is important for CTC survival, what is the downstream pathway of HA-CD44 in CTCs clustering?
6. It is quite surprising that HA-coated CTC clustering is prominent under shear stress. How does HA uniquely contribute under these conditions? What is the mechanism of HAS2-HA upregulation in CTCs? What upstream signals stimulate HAS2 to synthesize HA? or, metastatic TNBC cells constitutively express HAS2? Since clustering occurs very rapidly, in a minute, is it possible that shear stress can quickly induce HA secretion? In addition, given that HA-CD44 is involved in cell-cell interaction, this could be involved in cell-cell interactions in the primary tumor. What distinguishes HA in the blood stream from HA at the primary tumor site from HA? Additionally, how do cancer cells sense shear stress and respond to cluster formation via HA?
7. As authors stated in the Introduction, CD44 is expressed by many cell types, including immune cells. In Fig. 6, authors suggest that HA facilitates interactions between CTCs and non-CTCs. Any chance to work in heterotypic clusters with immune cells or CAFs? HA is also expressed by cancer cells and cancer-associated fibroblasts.
8. Non-metastatic cell lines, such as MCF7 and BT474, hardly express HAS2 (Fig. 1j), but they highly express EPCAM (extended Fig. 1l), showing almost no clustering (Fig. 1f). Given that EPCAM is also known to mediate cell-cell interactions, it is somewhat surprising that HA appears to have a stronger effect than EPCAM in promoting cluster formation. In addition, the HA-CD44 axis appears to act as a master regulator of CTC clustering, rather than functioning as an AJ-independent pathway. In Fig.2 and extended Fig.2, HAse treatment, HAS2 KD, or CD44 KO nearly abolishes all CTC clustering. This raises the question of whether HA-CD44 plays roles beyond just mediating cell-cell interactions.
9. The authors used 293FT cell line to investigate the role of HA-CD44 in TNBC clustering (Extended Fig.2). While it is understandable that they chose this line because it lacks both HAS2 and CD44, it is not a breast cancer cell line. It would be better if the authors could replicate these experiments in breast cancer cell line.
10. According to the literature, CD44 is known to be important for homophilic CTC clustering. However, Fig. 2c-d suggests CD44 function is impaired in the absence of HA. To directly test whether HA-CD44 interaction is crucial for TNBC CTC clustering, authors should consider block this interaction using a specific antagonist.

Reviewer #4

(Remarks to the Author)

Version 1:

Reviewer comments:

Reviewer #1

(Remarks to the Author)

In this resubmitted manuscript, the authors have responded conscientiously and directly to the concerns of the reviewers. The new Matrigel invasion results are interesting and support the interpretation of increased invasive metastasis. However, there seems to be a puzzle. This transfilter assay is thought to measure single-cell invasion, whereas the CTC clusters are

tighter. Do the authors have some idea or speculation on how to reconcile these apparent contradictions? For example, does HA increase both cell-cell adhesion in suspension and individual cell invasiveness when exposed to a matrix?

Reviewer #2

(Remarks to the Author)

The authors have addressed all my concerns.

Reviewer #3

(Remarks to the Author)

The authors have adequately addressed all reviewer comments in the revised manuscript.

Reviewer #4

(Remarks to the Author)

REVIEWER COMMENTS

Reviewer #1 (Remarks to the Author):

Overall impression:

This well-written manuscript provides an interesting and useful extension of prior research documenting the importance of CD44 in TNBC metastasis by providing compelling evidence that the main ligand of this cell surface receptor, hyaluronan (HA), plays an important role in cluster formation. While the evidence for CD44 and HA in cluster formation is extensive and convincing, the importance of this interaction in promoting metastasis is less clear.

Major points:

1. The authors state that “While similar numbers of both control and HAS2 KD cells reached the lungs upon injection, clear differences in metastatic lung signal are detectable from one week onwards (Fig. 5k-l and Extended Data Fig. 5h).” At the same time, the authors show that both control and HAS2 KD cells exhibit the same level of proliferation and Ki67 staining of the metastatic lungs was the same between LM2 control and HAS2 KD tumor cells (Extended Data Fig. 5i). Why would there be a difference in lung metastasis over time if cells’ ability to colonize the lungs and their proliferation rate was the same? Week 1-4 typically represents outgrowth of cells that have successfully colonized the lungs. Shouldn’t there be a difference in ability to colonize the lungs given that HAs-treated cells exhibit a 2-fold increase in ROS levels compared to control cells? In fact, the authors’ *in vitro* results show that “HA depletion decreased cell survival, with significant differences being detected at 16 hrs. At 24 hrs, only 20-25% of the HAs-treated or HAS2 KD LM2 cells were alive, compared to 65% of the control cells (Fig. 4e-f).”

Response:

We thank Reviewer 1 for raising this point and apologize for the confusion. We measured luciferase signals of each mouse immediately after tumor cell injection to ensure that the same number of tumor cells reached the lungs at time of injection. This number does not represent comparable lung colonization, as metastasis is an incredibly inefficient process. Studies have shown that most tumor cells that reach an organ fail to initiate macrometastases and instead either die or stay dormant (PMID: 9736035). As the Reviewer correctly pointed out, the *in vivo* cell proliferation rate is comparable between control and HAS2 KD cells (Extended Data Fig. 5f and 5k), indicating that the difference in metastasis is not a consequence of reduced outgrowth. Instead, initial lung colonization by HAS2 KD cells was likely impaired because HA-lacking cells are more susceptible to shear stress (revised Extended Data Fig. 4k-n).

We now present additional results to compare the lung colonization capability. We examined the lungs from mice 24 hrs after tail vein injection of LM2 control and HAS2 KD cells and detected a significantly lower number of HAS2 KD cells in the lungs compared to that of control (Fig. R1a and revised Extended Data Fig. 5a). Additionally, we performed *in vitro* invasion assay and found that HA-lacking cells have reduced invasion capability (Fig. R1b-c and revised Extended Data Fig. 5b). Taken together, these results reveal that HAS2 KD cells are likely more susceptible to the harmful effects of shear stress exposure, as the Reviewer suggested (see also Reviewer 3, point 5).

We have described the new results on page 16. It reads: “Based on our *in vitro* results, we hypothesized that HA-lacking CTCs may have reduced metastatic colonization ability as they are more susceptible to shear stress-induced cell death (Extended Data Fig. 4k-n). To test this hypothesis, we injected LM2 control and HAS2 KD cells via tail vein in female NSG mice and examined their lungs for the presence of GFP-positive tumor cells 24 hrs later. Indeed, HAS2 KD

tumor cells were present in fewer numbers in the lungs compared to control cells (Extended Data Fig. 5a). Accordingly, a Matrigel-based in vitro invasion assay revealed that HA-lacking single cells have reduced invasive capacity compared to clustered control cells (Extended Data Fig. 5b). Together, these results indicate that HA-mediated clustering protects tumor cells from shear-stress induced cell death and promotes invasion to enhance metastatic colonization and outgrowth.”

2. In Fig. 6f, the authors analyzed the number of CTCs originating from control and HAS2 KD cells in each mouse and observed a 4- to 12-fold reduction of CTCs originating from HAS2 KD cells. Why was a 4-week time-point chosen? Again, one would expect there to be a significant decrease in CTCs within 24 h based on the author’s in vitro data and previously published work (see PMID: 31485072, which the authors failed to cite in this manuscript). In addition, the authors need to rule out the possibility that even though HA is crucial for cluster formation, the final cell-cell contacts might be mediated by an additional cell-cell adhesion mechanism.

Response:

We thank the Reviewer for pointing out this important paper. We had cited it in an earlier version of the manuscript and accidentally deleted it. We have rectified this in the revised manuscript and now cite it as reference 71 in the discussion on page 19. It reads: “Highly aggressive TNBCs often exhibit mesenchymal features and downregulate epithelial AJ molecules to increase their invasive and migratory properties¹⁵⁻¹⁸. While this promotes dissemination, the lack of classical AJ-mediated interactions can prevent the formation of CTC clusters⁷¹.”

After tumor cell injection, CTCs are thought to get trapped in the capillaries and thus rapidly cleared from circulation (PMID: 26114035, 23166151). CTCs re-emerge in our system ~4 weeks post injection once lung lesions have been established and begin shedding cells. This is why we analyzed this specific time point. We tried to accommodate the Reviewer’s request and checked for CTCs 24 hrs after injection, but could not detect any CTCs. Of note, the publication referred by the Reviewer also quantifies CTC number only after a primary tumor has been fully established in a mammary fat pad spontaneous metastasis model. Our experimental setup shown in revised Fig. 5k-p is comparable to those in this published paper.

As mentioned in the response to the previous comment, we followed the suggestion of the Reviewer and now include data derived from lungs collected 24 hrs after tail vein injection, where we see reduced HAS2 KD tumor cells compared to control. Regarding the involvement of additional cell-cell adhesion molecules, we now include exciting new data showing that desmosomes help stabilize HA-based tumor cell clusters (see response to the next comment).

3. The authors state that “gap size is similar to the 10-20 nm reported for AJs; however, the cluster-forming metastatic cell lines lack the crucial AJ component E-cadherin.” Later, they also state that “LM2 control and HAse-treated cells showed similar numbers of clusters and single cells in the absence of shear stress (Fig. 4a and Extended Data Fig. 4a-c, “no shear stress”).” Although the evidence for the crucial involvement of HA in cluster formation is compelling, the authors have not ruled out the possibility that after HA-mediated cell attachment occurs, the final cell aggregates are held together most tightly by an alternative cell adhesive mechanism besides HA, even though E-cadherin is missing. Have the authors checked the expression level of other AJ components in LM2 cells? The data presented strongly suggests that another AJ component is involved in cell clustering. What happens if that component is knocked down? Can cells still form cell clusters effectively in response to shear stress?

Response:

We thank the Reviewer for raising the question concerning alternative cell adhesive mechanisms working in concert with HA. Two recent papers investigating CTCs showed an upregulation of the desmosomal cadherins Desmoglein-2 (DSG2) and Desmocollin-2 (DSC2), as well as Desmoplakin (DSP), which anchors these cadherins to intermediate filaments (PMID: 33431674, 34586853). Inspired by Reviewer 1’s question, we analyzed desmosomal proteins in our experimental system.

We found that all three proteins localize to mature, but not early cell-cell interaction sites (Fig. R2a-b, revised Fig. 4e, and revised Extended Data Fig. 4j). We further show that depletion of any of these desmosomal proteins (Fig. R2c and revised Extended Data Fig. 4i) does not abolish clustering per se, but affects their stability as clusters rapidly dissociate when experiencing higher shear stress (Fig. R2d-e and revised Fig. 4f-g). Together, this new set of experiments shows that desmosomes are established at the interaction site to further strengthen and stabilize HA-dependent clusters.

We added a new section named “Desmosomes stabilize HA-mediated clusters” on page 14 of the revised manuscript to present the desmosome data. It reads “The tight interactions observed in fully formed clusters (Fig. 1h-i, 4b-d and Extended Data Fig. 1k, 4g) prompted us to investigate whether HA-based clustering recruits additional components to strengthen the cell-cell connection during the maturation of the interaction site. Desmosomes are specialized intercellular junctions that connect to intermediate filaments, thereby providing strong mechanical coupling between neighboring cells⁵³. Desmoglein-2 (DSG2) and Desmocollin-2 (DSC2) are desmosomal cadherins that mediate adhesive binding between adjacent cells, while Desmoplakin (DSP) is the intracellular scaffold protein that anchors these cadherins to intermediate filament of the cytoskeleton. Interestingly, recent reports showed that high expression of DSG2 in CTCs increased prevalence of CTC clusters and metastasis⁵⁴ and that DSC2 and DSP was associated with increased cluster formation and CTC survival in breast and lung cancer cells⁵⁵.

These findings prompted us to investigate whether desmosomes are involved in HA-based clustering of tumor cells as LM2 cells express DSG2, DSC2, and DSP (Extended Data Fig. 4i). Assessing their localization in clustered LM2 cells, we found all three desmosome proteins localized to the fully established cell-cell interaction sites, indicating that they mechanically couple neighboring cells in mature clusters (Fig. 4e). In contrast, in early clustering interactions where

cell morphology is still largely round and cell membrane contacts are limited to a small area, only HA, but not DSG2, DSC2 or DSP, is enriched at the cell contact site (Extended Data Fig. 4j), suggesting that desmosomes are established after large areas of the cell membranes are brought into close proximity via HA-based interactions.

To determine whether desmosomes promote HA-based clustering by strengthening cell-cell interactions, we knocked down DSG2, DSC2, and DSP individually in LM2 cells (Extended Data Fig. 4i). Next, we performed cell clustering experiments with gradually increasing levels of rotational speed. All three KD cell lines formed clusters at 200rpm, but they rapidly dissociated at higher shear forces, indicating reduced stability compared to control cells (Fig. 4f-g). Of note, cluster stability in control cells was not as high as in Fig. 4a, as they started to fragment into smaller clusters at ~350 rpm (Fig. 4f). This suggests that while desmosome formation under shear stress is possible, it is less efficient than only challenging desmosome stability after they were fully established in the absence of shear stress. Taken together, these results suggest that the initial connection of cells through HA enables the subsequent assembly of desmosomes, which effectively lock in these interactions and mechanically stabilize the clusters.”

Furthermore, we also discuss the similarities between desmosomes formed following establishment of AJ- or HA-dependent cell-cell interactions on page 21: “The similarities further extend to the formation of desmosomes, which occurs only after initial membrane contacts have been established through AJs or HA. These similarities ultimately result in comparable structural features such as gap sizes between interacting cells and the stable connection of cells across a large membrane area^{39,40}.”

We thank Reviewer 1 for raising this question. The identification of desmosomes that stabilize the HA-mediated cell-cell interaction greatly strengthens our study.

Minor points:

1. An interesting observation is that freshly detached MCF7 cells fail to cluster when rotated on an orbital shaker. MCF7 cells express high levels of E-cadherin and form strong cell-cell contacts in the absence of shear force. If MCF7 cells are allowed to cluster and are then subjected to shear force, do they remain together as clusters? How strong is this interaction compared to HA-mediated cell clustering exhibited by LM2 cells? How do LM2 cells form clusters more readily in response to shear stress than MCF7 cells? Does HA act like a 'glue' making it more likely for colliding cells to interact with each other? Do E-cadherin based clusters take a long time to form whereas HA-CD44 form more quickly?

Response:

Indeed, the Reviewer's reasoning completely aligns with our own interpretation of the two clustering processes. The formation of adherens junctions is a complex interaction that involves multiple proteins, requiring hours to fully establish (PMID: 24659804). This is likely the reason why MCF7 cells fail to cluster when they briefly encounter each other in a highly volatile environment (revised Fig. 1f-g). In contrast, the simplicity of the HA-CD44 interaction enables dynamic interactions that can facilitate clustering even under shear stress, and subsequent assembly of desmosomes fully stabilize the interaction (see reply to previous comment). According to the Reviewer's question, we assessed the clustering capabilities of MCF7 cells under no shear stress conditions and found that MCF7 cells form clusters (Fig. R3 and revised Extended Data Fig. 4h). Once established, these clusters are stable and can withstand shear force (see also Reviewer 3, point 8).

We included this new result on page 13. It reads: "Examining the E-cadherin-positive MCF7 cell lines revealed that although they failed to cluster under shear stress (Fig. 1f-g), MCF7 cells can efficiently form clusters when left unperturbed in a low-attachment plate (Extended Data Fig. 4h). Subsequent addition of shear stress did not break apart the clusters, in line with the reported strength of AJ-dependent cell-cell interactions⁵¹ (Extended Data Fig. 4h). These results show that while HA is dispensable for TNBC cell cluster formation in the absence of shear stress, it is critically required to form functional tumor cell clusters that can withstand the shear stress encountered by CTC clusters in the bloodstream."

Fig. R3 MCF7 cells can form highly stable tumor cell clusters under no-shear stress conditions. (a) Representative picture of MCF7 clusters formed under no-shear stress (0 rpm) condition and tested for stability at higher rotational speeds (250 rpm and 350 rpm). (b) Quantification of pictures shown in a. (n=4). Cells were settled for 24 hrs in a low-attachment plate (0rpm) and then agitated sequentially at each speed for 15 min.

2. Increased HA intensity at cell-cell junctions may be due to the presence of more plasma membrane. For example, CAAX-GFP and GPI-GFP intensity is typically greater at cell-cell junctions. A better measure of HA intensity at cell-cell junctions would be to normalize the signal by the total amount of plasma membrane (i.e., HA/plasma membrane staining).

Response:

Following the Reviewer's excellent suggestion, we examined HA intensity compared to total amount of plasma membrane signal by staining cell membranes using Cellbrite® Fix Membrane. As seen in Fig. R4a-b and Extended Data Fig. 1o, HA is indeed significantly enriched at the interaction site.

We included this new data on page 7 as follows: "Importantly, quantification of HA compared to membrane staining further confirmed the enrichment of HA at cell junctions (Extended Data Fig. 1o)."

3. Have the authors performed an outlier test for Extended Fig. 3d? If the abnormally low data point for colchicine is removed, there is likely a significant increase in the number of single cells relative to control. Is the amount of CD44 on the plasma membrane altered when the microtubule network is dismantled? Perhaps the amount of HA on the plasma membrane is reduced?

Response:

As per the Reviewer's suggestion, we have performed outlier analysis using ROUT (Q=1%) in GraphPad Prism and no samples were marked for exclusion. Nonetheless, we decided to remove this particular replicate, as it also produced the outlier-like data point for number of clusters. Results comparing before and after removal of this replicate are shown in Fig. R5a vs. R5b). The new plots are included in revised Fig. 3e and revised Extended Data Fig. 3d).

We thank the Reviewer for raising the question concerning CD44 and HA levels in treated cells, as it uncovered a surprising phenotype. Both CD44 and surface-bound HA were almost completely absent in Latrunculin A-treated cells, whereas colchicine didn't show any effect (Fig. R5c and revised Extended Data Fig. 3c). We have updated our description to accommodate this new finding on page 10 as follows: "Treatment of LM2 cells with the actin depolymerizing drug

Latrunculin A, but not the microtubule polymerization inhibitor colchicine, eliminated protrusions⁴⁴ and increased cell size⁴⁵ (Extended Data Fig. 3c). In addition, cell surface staining of CD44, and especially HA, was markedly reduced in Latrunculin A treated cells (Extended Data Fig. 3c) suggesting that actin is required to properly position CD44, and thereby also HA at the cell surface. Consistent with HA being required for TNBC tumor cell clustering (Fig. 2 and Extended Data Fig. 2), Latrunculin A treatment abolished cluster formation, whereas clustering remained largely unaffected by colchicine (Fig. 3e and Extended Data Fig. 3d). Together, these results suggest that actin is essential for the formation of tumor cell clusters, likely through the formation of cellular protrusions that help organize HA localization (Fig. 2j and 3a).”

4. Line 84: Has hyaluronan been identified previously among ECM components reported to be upregulated in CTCs?

Response:

To our knowledge, hyaluronan (HA) has not been previously identified to be upregulated in CTCs. This is likely due to multiple factors. First, there is no direct measure for HA in RNA-seq datasets as it is a polysaccharide. Second, HA levels are dynamically regulated by multiple synthases and degradation factors called hyaluronidases. These proteins are variably expressed across tissues, making HA level prediction difficult. Third, HA synthases are not structural ECM components and thus may not necessarily be classified as such in bioinformatic analyses. Lastly, CTCs are an extremely rare population and may become damaged during isolation. These challenges might have limited the discovery of HA's association with CTCs.

5. Lines 236-237: The evidence presented does not show conclusively that “actin filaments provide the protrusive force for the formation of these structures” but instead that actin is necessary for their maintenance.

Response:

We agree with the Reviewer's comment and have removed this statement as Latrunculin A treatment cannot distinguish whether maintenance or formation of the protrusions is affected. In addition, our finding that CD44 and HA cell surface localization is perturbed by the treatment makes it likely that the failure to cluster is a direct consequence of the altered HA localization and not necessarily the perturbed protrusions. On page 10, the revised manuscript reads as follows: "Consistent with HA being required for TNBC tumor cell clustering (Fig. 2 and Extended Data Fig. 2), Latrunculin A treatment abolished cluster formation, whereas clustering remained largely unaffected by colchicine (Fig. 3e and Extended Data Fig. 3d). Together, these results suggest that actin is essential for the formation of tumor cell clusters, likely through the formation of cellular protrusions that help organize HA localization (Fig. 2j and 3a)."

Wording:

1. Line 75: Typo: emboli rather than embolies

Response:

We now use the correct word "emboli".

2. Line 232: The "remarkable overlap" is not surprising because protrusions are normally filled with actin, so actin should overlap with any cell surface or matrix component on the outside of protrusions.

Response:

This wording was chosen in comparison to the microtubule staining. We rewrote this statement in line with the Reviewer's comment to avoid making the overlap sound like something extraordinary. The revised manuscript now reads on page 10: "Actin staining showed a clear overlap with the CD44-marked protrusions (Fig. 3d), whereas tubulin localized just beneath the cell membrane (Extended Data Fig. 3b)."

3. Line 482: The wording appears too general: the non-tumor cells also need to be CD44-expressing.

Response:

We agree with the Reviewer and now emphasize the requirement of the HA receptor CD44. The sentence read: "Using TNBC patient samples and mouse models, we also show that HA can serve as a landing pad to enable integration of non-tumor cells that express the HA receptor CD44 into heterotypic CTC clusters." in the Line on page 20 that corresponds to Line 482 of the initial manuscript.

Reviewer #2 (Remarks to the Author):

This manuscript by Bobkov et al, identified a previously unrecognized HA-mediated mechanism that enables stable CTC cluster formation in triple-negative breast cancer (TNBC). The study is well designed, integrating both in vitro and in vivo assays with human samples and mouse models. The data are convincing, and the findings are both novel and highly clinic relevant. The results expand the current paradigm of CTC cluster biology and may provide new possibilities for treating metastatic TNBC patients. Some comments are shown below to improve the manuscript:

Major comments:

1. In most of the images showing cell clustering (e.g., Fig. 1f, Fig. 2b, 2g, etc.), the non-clustered groups have less cells than the clustered groups. It would be better to show full images of the entire culture to demonstrate that each group started with a similar number of cells.

Response:

We fully acknowledge the concern raised by the Reviewer and explored options to add the full overview images. As can be seen in Figure R6a, the full image is too large to allow visualization of individual cells. To accommodate the Reviewer's request, we now show a large field of cells to reflect that equal amount of cells were used (Fig. R6b and Extended Data Fig. 2a).

2. Please carefully double-check all image labels. For example, in Fig. 5k, the signal based on the bar shown should be between 150–2,000 on Day 0, but the y-axis is labeled as 10^4 in Fig.5i. In addition, the Week 4 raw signal appears to be between 2,000–20,000(Fig.5K), so it is unclear how the “signal (relative to Day 0)” can be $\sim 1,000$ (fig.5i).

Response:

We thank the Reviewer for pointing this out. The IVIS picture shown in original Fig. 5k are not directly comparable to the quantified tumor burden shown in original Fig.5i. The representative images for d0 (Fig. R7a and revised Fig. 5k) include a heatmap to visualize bioluminescence signal intensity per pixel. The scale bars of 35-600 (Day 0) and 200-2k (Week 4) depict the range

of minimum and maximum signal intensity per pixel. In the revised Fig. 5l, combined total radiance signal of the lungs is shown. Therefore, the numbers in the bar plots in revised Fig. 5l reflect the “total normalized signal” and are thus not directly comparable to the “signal/pixel” shown in the heatmap in revised Fig. 5k. Additionally, we accidentally showed representative images of mice from the pilot experiment in Fig. 5k and have now replaced them with the ones used in the final experiment. The revised Fig. 5k and 5l are shown in Fig. R7.

3. The inclusion of patient-derived CTCs is a strength, but the sample size is not specified. In addition, including CTCs from other breast cancer subtypes for comparison, if feasible, would further enhance the translational relevance of the findings.

Response:

According to the Reviewer’s request, the updated manuscript now includes the sample size and reads on page 19: “We next analyzed HA localization in blood specimens from five TNBC patients. CTCs from patient blood were identified using a cocktail of the gold standard CTC markers (Pan-Cytokeratin, EpCAM and EGFR). CTC clusters were detected in three of the TNBC patients. These CTCs displayed a prominent HA coat, and a clear HA enrichment was found at CTC-CTC contact sites, both in clusters where all CTCs produce HA (Extended Data Fig. 6e) or when only one CTC provides the HA necessary for the interaction (Fig. 6j). One TNBC patient had HA-coated CTCs, but no CTC clusters were detected, and the last TNBC patient only had single CTCs without an HA coat. These results reiterate the importance of HA in TNBC CTC clustering and also show that HA-coated CTCs are capable of recruiting HA-lacking CTCs into CTC clusters.”

We further analyzed CTC clusters from blood specimens of four ER⁺/PR⁺/HER2⁻ patients using previously banked slides known to contain CTC clusters. As expected, none of the CTCs showed cell-surface HA staining, indicating that these clusters likely form through a different mechanism. We predict that they are mediated by E-cadherin. We elected to omit these data from the final manuscript, since this study does not otherwise focus on E-cadherin-dependent clustering and we were unable to stain for E-cadherin due to limited availability of patient slides. We chose not to include the non-TNBC patient data in order to keep the study focused and

minimize potential confusion for readers. Nonetheless, we would be happy to include these data in the revised manuscript if requested.

4. In addition to the RT-PCR data, the knockdown efficiency of HAS2 should be validated by Western blot. Since the pooled population contain residual HAS2, and the remaining HA can mediate cell clustering in HAS2 knockdown cells (as shown in Extended Data Fig. 5I). Providing this additional validation would strengthen the conclusions regarding HAS2's role.

Response:

Following the Reviewer's suggestion, we now include a WB showing the knockdown efficiency of HAS2 on the protein level (Fig. R8 and revised Extended Data Fig. 2c) to supplement our original qPCR and HA IF data (revised Extended Data Fig. 2b,d).

5. In the conclusion, it is stated that: "The formation of this coat is likely a consequence of HA's unique mode of production. HA is synthesized at the plasma membrane, and while the growing HA chain extrudes out into the extracellular space, it is immediately captured and bound by its receptor CD44. This enables the formation of an HA-rich pericellular coat." Could the authors provide citation to support this statement? Otherwise, additional experiments would be needed to validate it.

Response:

We agree with the Reviewer that a citation is warranted at this place and now cite a comprehensive review about HA synthases that describes the unique production method of this major ECM component in detail (PMID: 17981795 – Reference 32 in revised manuscript) and an additional review that examines the Golgi-driven secretion of other ECM macromolecules (PMID: 35023559 – Reference 70 in revised manuscript). The notion that HA is bound by CD44 at the cell surface is further supported by the fact that cell-membrane associated HA is lost in CD44 KO cells (Fig. R9a and revised Extended Data Fig. 2h, j).

6. It is known that CTC clustering can promote cell survival, and HA has also been reported to support cell survival. Therefore, the cell survival data may be considered to remove or move to the supplemental data, as it is not central to the focus of the manuscript. In addition, the reactive oxygen species (ROS) data is weak. If these data are kept, the authors could strengthen this aspect by performing rescue experiments—for example, testing whether antioxidants can prevent the death of Hase-treated cells.

Response:

We thank the Reviewer for these suggestions. We included this experiment to show that specifically the HA-mediated clusters can enhance cell survival. As per the Reviewer's suggestion, both the cell survival and ROS experiment data have now been moved to the supplemental data (Fig. R10a and revised Extended Data Fig. 4k-n). Additionally, we have performed the suggested rescue experiment and found that the antioxidant N-acetylcysteine (NAC) can partially rescue the cell death phenotype of both clustered control and single HAse-treated cells (Fig. R10b and revised Extended Data Fig. 4n), suggesting that HA-mediated clusters partially protect tumor cells against shear stress-induced cell death.

The updated text in the revised manuscript can be found on page 15 and reads “Since cell clustering can limit ROS production⁵⁹, we tested whether oxidative stress contributes to the increased shear stress-induced cell death in HA-lacking cells (Extended Data Fig. 4k-l). Compared to untreated wild type cells, shear stress exposure elevated ROS levels in both control and HAse-treated LM2 cells at 12 hrs (Extended Data Fig. 4m), a timepoint preceding extensive cell death of HAse treated cells (Extended Data Fig. 4k). However, the ROS levels were more than two-fold higher in single HAse-treated cells compared to clustered control (Extended Data Fig. 4m), suggesting that HA-mediated clustering promotes CTC survival by limiting shear stress-induced ROS production. Addition of the antioxidant N-acetylcysteine (NAC) to repress ROS moderately rescued cell death of HAse-treated single cells (Extended Data Fig. 4n). suggesting that HA-mediated clustering at least partially shields tumor cells from shear stress-induced cell death.”

Minor comments:

1. Since SKBR3 cells are used in Fig. 1g, it would be more appropriate for the representative image in Fig. 1f to also show SKBR3 cells, rather than BT474 cells. This would help maintain consistency between the panels.

Response:

We thank the Reviewer for catching this. We have adapted the Figures according to the Reviewer's suggestion.

2. In the text, it is stated that “CTC clusters were detected in all 6 control mice”. However, only 5 mice are shown in fig.5h& 5i.

Response:

We thank the Reviewer for catching this. The discrepancy between the text and Figure was a typo and has been corrected.

3. Since the tumor cells were labeled with GFP, it is unclear whether the GFP signal will be preserved during IF staining, especially given that HA is also detected using a green fluorophore. Could the authors clarify how these signals are distinguished in the images?

Response:

We apologize for the confusion; HA was stained in the far-red channel, and not in the green channel. The green color was only chosen as a pseudo-color. For presentation purposes, we kept all HA signals in green throughout the manuscript. To avoid any confusion, we have now edited the figure legends and also included a clarification in the method section. The Figure legends read “For visualization, HA staining was assigned a green pseudo-color. GFP signal is shown in grey in the Extended Data Fig. 5/6”. We have further indicated wherever a pseudo red/green color was used (if imaged cells express RFP or GFP respectively) by changing the description from “stained for HA (green)” to “stained for HA (green pseudo color)” to prevent future confusion.

Reviewer #3 (Remarks to the Author):

In this manuscript, Bobkov and Patel et al. propose a novel role for hyaluronan in circulating tumor cell (CTC) clustering during TNBC metastasis. They show that CTCs express HAS2, which synthesizes HA, and that HA interacts with CD44 on cancer cells to function as adherent junctions, which is very interesting and novel. Overall, the data are clear and well presented. However, some of the results sections are somewhat lengthy and would be better reorganizing or moving parts into the discussion. It is intriguing that HA appears to show such a strong effect as a mediator of cell-cell interactions, the data overall support this hypothesis well. Detailed comments are as below:

1. HA is a well-known serum marker for several cancers, including breast, prostate, and gastrointestinal cancers. Elevated serum HA levels have been correlated with a worse prognosis and increase risk of metastasis, which supports the findings of this study. However, this also indicates that HA is already abundant in the bloodstream of patients with cancer or tumor-bearing mice, suggesting that CTCs may not be the primary source of HA. Moreover, HA is highly accumulated in the primary tumor, raising the possibility that HA-coated cancer cells could detach from the primary tumor, intravasate, and form clusters. According to this study, endogenous HAS2 expression is crucial for HA-mediated CTC clustering (HAS2 overexpression and knockdown experiments). In the context of rich HA in the bloodstream or culture media, CTCs can form clusters independently of HAS2 expression? Can you also compare HAS2 expression or HA intensity in LM2 clusters compared to LM2 single cells?

Response:

We thank the Reviewer for their recognition of our work. We have also edited the manuscript to make it more concise.

Following the Reviewer’s suggestion, we tested whether exogenous HA in the culture media can facilitate tumor cell clustering and found that high molecular weight (HMW; 750-1000kDa), but not low molecular weight (LMW; 130-150kDa) HA can rescue the clustering defect of LM2 HAS2 KD cells in a dosage dependent manner (Fig. R11 and revised Extended Data Fig. 2g).

This new data is shown on page 7-8 of the revised manuscript and reads “HA molecules can exist in different sizes and can elicit distinct cellular responses. To assess what form of HA facilitates tumor cell clustering, we treated LM2 HAS2 KD cells for 1 hr with increasing

concentrations of two different sizes of HA molecules, a low molecular weight (130-150 kDa) HA and a high molecular weight (750-1000 kDa) HA. Incubation with exogenous high molecular weight HA reconstituted clustering capabilities of LM2 HAS2 KD cells in a dose-dependent manner, whereas low molecular weight HA failed to rescue clustering (Extended Data Fig. 2g). These results are consistent with HAS2's function in producing HA molecules of higher molecular weight⁴¹ and suggest that large HA molecules are required to mediate tumor cell clustering, likely by providing enough interaction sites for HA receptors on two or more tumor cells to stably interact with the same HA moiety.”

While these results suggest that high abundance of exogenous HA of the appropriate size can mediate CTC clustering, such conditions are unlikely to occur *in vivo*. HA in the tumor microenvironment is thought to be a mixture of various sizes and is likely skewed towards LMW HA due to increased degradation by hyaluronidases and the resulting pro-tumorigenic effects (PMID: 31134064, 26106384). Although HA levels can be elevated in the bloodstream of breast cancer patients, they generally remain low in absolute concentration (PMID: 2394505, 26686298, 26082778). Moreover, serum LMW HA (<50 kDa), but not total HA, was found to be correlated with breast cancer metastasis (PMID: 25550464), suggesting that elevated serum HA is insufficient to facilitate CTC clustering. Together, these factors help explain why HAS2 KD tumor cells in our *in vivo* experiments couldn't compensate for their lack of HA by utilizing stromal HA to form CTC clusters.

Regarding the last question, the HA intensity is not altered in clustered and single LM2 cells, but the localization of HA shifts during the maturation of the cell-cell interaction, where it transforms from an even coating of the cell surface (Fig. R12a and revised Fig. 6c) to enrichment at the interaction sites (Fig. R12b and revised Extended Data Fig. 6g).

2. Metastatic tumors often lose adherens junction expression as they undergo epithelial-to-mesenchymal transition. This transition is a very important process in metastasis. In this study, EpCAM-negative cancer cells show high HAS2 expression, whereas EpCAM-positive cancer cells exhibit little to no HAS2 expression (Fig. 1j and extended Fig.1l). Is there any evidence linking the HAS2 or HAS2-HA -CD44 axis to epithelial-to-mesenchymal phenotypic changes? And could this transition also be associated with CTC clustering or cell-cell interactions?

Response:

We thank the Reviewer for raising the important question whether HAS2 is linked to EMT phenotypic changes. The connection between HAS2 and EMT has indeed been documented by multiple publications, such as HAS2 being required for TGF β induced EMT (PMID: 23108409), a ZEB1/HAS2 positive feedback loop that promotes EMT in breast cancer (PMID: 28086235) and our previous work showing that depletion of HAS2 abolishes the effect of CD44s on promoting the mesenchymal phenotype (PMID: 28533273). EMT traits highly correlate with HA-based CTC clustering (revised Fig. 1c), as cells undergoing EMT downregulate cell-cell interaction molecules and instead upregulate adhesion molecules focused on cell-ECM interactions (PMID: 24556840, 15688013, 22778267, 23165231). We have now included the connection of HAS2/HA to EMT as well as their connection to CTC clustering in Discussion. In our study, we measured E-Cadherin (Extended Data Fig. 1l) and interpreted the “EpCAM” mentioned in the comment as a referral to E-cadherin.

3. While this study primarily focused on the TNBC metastasis, it would be important to clarify whether similar observations are found in other aggressive or malignant cancer types.

Response:

We thank the Reviewer for this great suggestion. We assessed the clustering capabilities of three different cell lines from other aggressive cancer types: the pancreatic PANC-1, the prostate PC3, and the glioblastoma DAOY. All three cell lines exhibit similar HAS2 and CD44 expression (Fig. R13a and revised Extended Data Fig. 2s). Consistent with the HA-based clustering mechanism identified in TNBC tumor cells, all three lines rapidly form clusters when subjected to our cluster assay, and Hase treatment abolishes cluster formation (Fig. R13b-c and revised Fig. 2l-m). These results show that the HA-based clustering mechanism is a feature shared by other aggressive and malignant cancer types.

We included this new data on page 9. It reads “To further generalize our findings in other cancer cell types, we performed the clustering assay using the prostate PC-3, the pancreatic PANC-1, and the glioblastoma DAOY cell lines. All three cell lines express HAS2 and CD44 at similar levels (Extended Data Fig. 2s), formed clusters under shear stress, and Hase treatment abolished clustering Fig. 2l-m). Together, these results support the notion that the HA-based clustering mechanism is employed by many metastatic cancer types in addition to TNBC tumor cells.”

4. HA exists in both high-molecular weight and low-molecular weight forms, which are known to play distinct roles in pathophysiology. The authors should determine the size of HA to better understand the novel function of HA.

Response:

Please see our response to Reviewer 3 Comment 1. We determined the size of HA and found high-molecular weight HA to be required for the clustering process.

5. The authors suggest that tumor cells express HAS, which synthesize HA, and that this HA interacts with CD44 on cancer cells, forming a kind of autocrine loop. This axis mediates cell-cell interactions and promotes CTC clustering. The HA-CD44 interaction has been extensively studied in cancer biology. Do HA-coated CTC clusters also contribute to extravasation, cancer cell migration, or invasion, at later step of metastasis? In vivo data suggest that HA-mediated CTC clustering plays an important role in TNBC lung metastasis (Fig.5). Given that HA-CD44 is important for CTC survival, what is the downstream pathway of HA-CD44 in CTCs clustering?

Response:

We thank the Reviewer for this question. Accordingly, we performed an *in vitro* invasion assay to determine whether HA-coated CTC clusters also contribute to invasion. Our results shown in Fig. R1b and revised Extended Data Fig. 5b are now included on page 14. It reads: “Additionally, a Matrigel-based *in vitro* invasion assay revealed that HA-lacking single cells have reduced invasive capacity compared to clustered control cells (Extended Data Fig. 5b).”

Additionally, we performed a 24 hrs tail vein experiment to examine differences in lung colonization between pre-aggregated LM2 control and HAS2 KD cells. The resulting increased colonization by control cells further suggests that HA-coated clusters have enhanced ability to survive shear stress, extravasate and/or colonize metastatic sites. The results are shown in response to Reviewer 1 Comment 1.

Regarding the downstream signaling pathways induced by CD44-HA interaction, we and others have shown that CD44-HA interaction activates the PI3K-Akt signaling pathway (PMID: 28533273, 33024087, 11408591, 12145277). This survival advantage of HA-dependent CTC clusters is further supported by our results shown in the revised Extended Data Fig. 4k-n.

6. It is quite surprising that HA-coated CTC clustering is prominent under shear stress. How does HA uniquely contribute under these conditions? What is the mechanism of HAS2-HA upregulation in CTCs? What upstream signals stimulate HAS2 to synthesize HA? or, metastatic TNBC cells constitutively express HAS2? Since clustering occurs very rapidly, in a minute, is it possible that shear stress can quickly induce HA secretion? In addition, given that HA-CD44 is involved in cell-cell interaction, this could be involved in cell-cell interactions in the primary tumor. What distinguishes HA in the blood stream from HA at the primary tumor site from HA? Additionally, how do cancer cells sense shear stress and respond to cluster formation via HA?

Response:

These are a set of great questions. We found that HAS2 is upregulated in highly metastatic TNBC cells and constitutively expressed (revised Fig. 1b, 1j), which enables TNBC CTCs to form HA-dependent clusters. Our previous study (PMID: 28533273) showed that CD44/HA interaction elicits a positive-feedback loop that stimulates PI3K/Akt, resulting in inhibition of FOXO1, a transcription repressor of HAS2. In turn, this results in upregulation of HAS2, producing HA. We agree with the Reviewer that HA-CD44-mediated cell-cell interactions can occur in the primary tumor, although as discussed in our response to Reviewer 3, Comment 1, the molecular weight of the HA present in the tumor microenvironment and the bloodstream is likely of LMW and thus not able to induce tumor cell clustering. Instead, our results suggest that tumor-derived HA is key for the clustering process (revised Fig. 5, 6, and revised Extended Data Fig. 5, 6). How cancer cells sense shear stress and respond to cluster formation via HA is an excellent question. As it is out of the scope of current study, we will investigate it in the future.

7. As authors stated in the Introduction, CD44 is expressed by many cell types, including immune cells. In Fig. 6, authors suggest that HA facilitates interactions between CTCs and non-CTCs. Any chance to work in heterotypic clusters with immune cells or CAFs? HA is also expressed by cancer cells and cancer-associated fibroblasts.

Response:

As shown in revised Fig. 6k-n and revised Extended data Fig. 6f-i, we show that immune cells can indeed be recruited into heterotypic CTC clusters. Importantly, CTC-bound HA molecules translocate to the interaction site with the HA-lacking immune cells, supporting the notion that HA from CTCs promotes the interaction between CTCs and immune cells. We thank the Reviewer for bringing up CAFs, as they have been shown to form heterotypic clusters with CTCs (PMID: 34216317), and we now cite it as reference 8 in the revised manuscript. It reads on page 3: "Metastatic spreading of tumor cells to secondary sites remains the major cause of cancer-related deaths. Circulating tumor cells (CTCs) represent an intermediate stage of metastasis and travel through the blood as single cells or as clusters. The latter can be composed solely of CTCs, or exist as heterotypic clusters that include other cell types, such as white blood cells¹⁻⁴, platelets^{5,6}, or cancer-associated fibroblasts^{7,8}."

8. Non-metastatic cell lines, such as MCF7 and BT474, hardly express HAS2 (Fig. 1j), but they highly express EPCAM (extended Fig. 1l), showing almost no clustering (Fig. 1f). Given that EPCAM is also known to mediate cell-cell interactions, it is somewhat surprising that HA appears to have a stronger effect than EPCAM in promoting cluster formation. In addition, the HA-CD44 axis appears to act as a master regulator of CTC clustering, rather than functioning as an AJ-independent pathway. In Fig.2 and extended Fig.2, Hase treatment, HAS2 KD, or CD44 KO nearly abolishes all CTC clustering. This raises the question of whether HA-CD44 plays roles beyond just mediating cell-cell interactions.

Response:

We believed the Reviewer meant E-cadherin and not EpCAM, as we did not discuss EpCAM in the manuscript. The Reviewer is correct that E-Cadherin can facilitate cell clustering (PMID: 26831077). Please see also our response to Reviewer 1, Minor point 1. We have now streamlined our description and added new data to support our conclusion. We show that the E-cadherin-positive MCF7 cells fail to form clusters under shear stress but can form clusters under no-stress conditions (Fig. R3 and revised Extended Data Fig. 4h). Once formed, these clusters are highly stable and capable of withstanding shear stress (Fig. R3 and revised Extended Data Fig. 4h).

9. The authors used 293FT cell line to investigate the role of HA-CD44 in TNBC clustering (Extended Fig.2). While it is understandable that they chose this line because it lacks both HAS2 and CD44, it is not a breast cancer cell line. It would be better if the authors could replicate these experiments in breast cancer cell line.

Response:

We thank the Reviewer for their comment. The purpose of using 293FT cells was to demonstrate that a CD44-negative and HAS2-negative non-clustering cell line can acquire clustering ability upon introduction of CD44 and HAS2 expression, and that the CD44-R41A mutant, which cannot bind HA, fails to confer this clustering ability (revised Fig. 2h-j and revised Extended Data Fig. 2o-r). Similarly, we also transformed the non-clustering TNBC line SUM159 into a clustering one by inducing the formation of an HA coat through the ectopic expression of HAS2. In line with the Reviewer’s suggestion, we updated the title of the respective paragraph on page 8 from “HA-CD44 interaction is sufficient to induce TNBC cell clustering” to “HA-CD44 interaction is sufficient to induce tumor cell clustering”. This broader title further allows inclusion of the newly added data showing that non-breast cancer metastatic cell lines can cluster through HA (Fig. R13 and revised Fig. 2l-m) into the paragraph (see Reviewer 3, comment 3).

10. According to the literature, CD44 is known to be important for homophilic CTC clustering. However, Fig. 2c-d suggests CD44 function is impaired in the absence of HA. To directly test whether HA-CD44 interaction is crucial for TNBC CTC clustering, authors should consider block this interaction using a specific antagonist.

Response:

The Reviewer is correct that CD44 has been reported to be important for homophilic CTC clustering (PMID: 30361447). We recapitulated previous results showing CD44 is required for clustering in a no-shear stress setting. We specifically addressed the differences between our results and the earlier paper in Fig. 4. In short, homophilic CD44 interaction alone can facilitate loose tumor cell aggregation under no-stress conditions. In the absence of HA, however, these clusters are not functional as they rapidly dissociate under physiological levels of shear stress (revised Fig. 4a-d and revised Extended Data Fig. 4a-g). The presence of HA enables tumor cell clusters to withstand shear stress CTC clusters encounter in the bloodstream.

To show HA-CD44 interaction is crucial, we performed two sets of experiments. First, in line with the Reviewer’s comment, we used a CD44 antibody as a specific antagonist and showed that it abolished tumor cell clustering (Fig. R14a and revised Fig. 2k). Moreover, we used the CD44 R41A mutant that prevents its interaction with HA and demonstrated that the HA-CD44 interaction is crucial for cell clustering (revised Fig. 2h-j and revised Extended Data Fig. 2o-r). We thank the Reviewer for their comments.

Reviewer #4, ECR co-reviewing with Reviewer #1(Remarks to the Author): I co-reviewed this manuscript with one of the reviewers who provided the listed reports. This is part of the Nature Communications initiative to facilitate training in peer review and to provide appropriate recognition for Early Career Researchers who co-review manuscripts.

REVIEWER COMMENTS

Reviewer #1 (Remarks to the Author):

In this resubmitted manuscript, the authors have responded conscientiously and directly to the concerns of the reviewers. The new Matrigel invasion results are interesting and support the interpretation of increased invasive metastasis. However, there seems to be a puzzle. This transfilter assay is thought to measure single-cell invasion, whereas the CTC clusters are tighter. Do the authors have some idea or speculation on how to reconcile these apparent contradictions? For example, does HA increase both cell-cell adhesion in suspension and individual cell invasiveness when exposed to a matrix?

Response:

We thank the Reviewer for this thoughtful comment. Previous studies have shown that TNBC cell clusters can traverse capillary-sized constrictions of 5 μm by dynamically rearranging the cluster structure. Tumor cells within the original cluster remain connected while passing through the constriction in a single-file fashion, and reform into clusters upon exiting. This work was discussed in the Discussion section of the manuscript (Ref. 74). Although the pore size of transwell would primarily permit the passage of single cells or very small clusters, HA-mediated clusters may similarly transverse the transwell in a single-file configuration. While unlikely, we cannot exclude the possibility that clustered cells migrate through adjacent pores as individual cells, which would suggest that HA directly enhances cell invasion. Nonetheless, even under this scenario, the findings remain consistent with a contribution of HA to promote metastatic processes.

Reviewer #2 (Remarks to the Author):

The authors have addressed all my concerns.

Reviewer #3 (Remarks to the Author):

The authors have adequately addressed all reviewer comments in the revised manuscript.

Reviewer #4 (Remarks to the Author):
